# Learning Safe Control via On-the-Fly Bandit Exploration

Alexandre Capone [1]   Ryan Cosner [2]   Aaron Ames [2]   Sandra Hirche [3]

## Abstract

Control tasks with safety requirements under high levels of model uncertainty are increasingly common. Machine learning techniques are frequently used to address such tasks, typically by leveraging model error bounds to specify robust constraint-based safety filters. However, if the learned model uncertainty is very high, the corresponding filters are potentially invalid, meaning no control input satisfies the constraints imposed by the safety filter. While most works address this issue by assuming some form of safe backup controller, ours tackles it by collecting additional data on the fly using a Gaussian process bandit-type algorithm. We combine a control barrier function with a learned model to specify a robust certificate that ensures safety if feasible. Whenever infeasibility occurs, we leverage the control barrier function to guide exploration, ensuring the collected data contributes toward the closed-loop system safety. By combining a safety filter with exploration in this manner, our method provably achieves safety in a setting that allows for a zero-mean prior dynamics model, without requiring a backup controller. To the best of our knowledge, it is the first safe learning-based control method that achieves this.

## 1. Introduction

With the growing proliferation of robotics in safety-critical fields, e.g., autonomous vehicles, medical robotics, and aerospace systems, the need for methods that ensure the safety of systems and their users has become paramount. In control tasks, guaranteeing safety has become synony-mous with guaranteeing that the system always satisfies a pre-specified set of constraints. While there exist various tools to achieve safety whenever the system is known perfectly (Agrawal & Sreenath, 2017; Ames et al., 2017; Bansal et al., 2017; Wabersich & Zeilinger, 2018; Zeng et al., 2021), doing so if the dynamics have to be learned poses a significantly more difficult challenge.

To address safety in a control setting, existing approaches typically attempt to bound the model error and assess how its worst-case estimate affects safety. To estimate the model error, frequently used frameworks include ensembles of neural networks (Chua et al., 2018; Curi et al., 2020) or Gaussian processes (GPs) (Capone et al., 2022; Rodriguez et al., 2021). The impact of the model error on safety is then frequently estimated by predicting the worst-case trajectory multiple steps into the future (Akametalu et al., 2014; Hewing et al., 2020; Koller et al., 2018). Alternatively, safety filters attempt to condense the long-term impact of a control input on safety (Ao et al., 2025; Berkenkamp et al., 2017; Castañeda et al., 2021; Cheng et al., 2019; Jagtap et al., 2020). A crucial drawback of most of these methods is that they assume the model uncertainty is always small enough for the conservative safety constraints to be feasible. In other words, if the uncertainty becomes too large, the underlying methods are no longer applicable.

Motivated by the aforementioned challenges, we present a learning-based control algorithm that ensures safety by exploring on the fly. We use a Gaussian process model to estimate model uncertainty, which we combine with a control barrier function to compute a robust safety filter. Whenever our safety filter becomes infeasible, we employ the control barrier function to guide exploration in a manner that is informative for safety. Formally, this is done by leveraging the lower confidence bound of the certificate constraint function for certifiably safe control inputs, and the upper confidence bound for exploration. Our approach draws heaviliy from ideas from Bayesian optimization, where exploration is done in a similar manner. Overall, our main contributions are as follows:

1. We present the first safe control algorithm that addresses infeasible safety filters by exploring online using a Gaussian process bandit algorithm.

*Equal contribution  [1]Robotics Institute, Carnegie Mellon University, Pittsburgh, PA, USA [2] Department of Mechanical and Civil Engineering , California Institute of Technology, Pasadena, CA, USA [3]TUM School of Computation, Information and Technology, Technical University of Munich, Munich, Germany. Correspondence to: Alexandre Capone <acapone2@andrew.cmu.edu>.

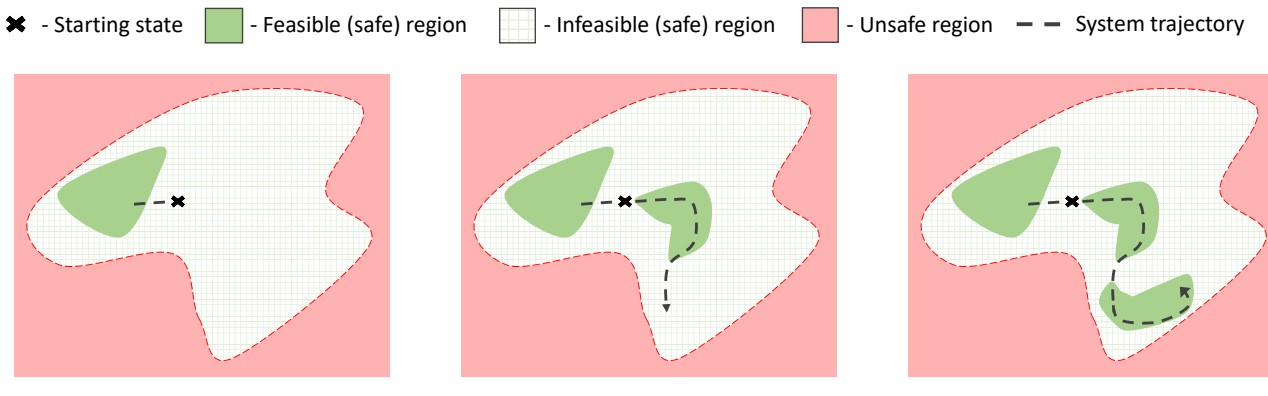

✖ - Starting state    🟩 - Feasible (safe) region    ⬜ - Infeasible (safe) region    🟥 - Unsafe region    – – System trajectory

**(a)** System state exits the region where the safety filter is feasible.

**(b)** Our algorithm computes exploratory control inputs using a bandit-type technique.

**(c)** Theorem 3.2 guarantees that the safety filter always becomes feasible before the boundary of the safe set is reached.

*Figure 1.* Illustration of our method (Algorithm 1) and theoretical result (Theorem 3.2).

2. We rigorously analyze our approach and discuss our main theorem, which shows the interdependence between model uncertainty, sampling time, and safety.

3. We showcase our approach using two numerical simulations where we achieve safety using a control barrier function, a dynamics model with a zero-mean prior, and no prior measurements. To the best of our knowledge, this setting cannot be addressed by current state-of-the-art methods.

The remainder of this paper is structured as follows. In Section 2 we describe the problem setting considered in this paper, together with some background on control barrier functions and Gaussian processes. Our main contribution, which includes an exploration-driven safe control algorithm and safety guarantees, is presented in Section 3. In Section 5, we apply our method using numerical simulations of cruise control and a quadcopter. We finalize the paper with some conclusions in Section 7.

## 2. Background and Problem Setting

We consider a system that is affine in the control inputs

$$\dot{x} = f(x) + g(x)u \qquad (1)$$

where $x \in \mathcal{X} \subseteq \mathbb{R}^n$ and $u \in \mathcal{U} \subseteq \mathbb{R}^m$, and $f : \mathbb{R}^n \to \mathbb{R}^n$ and $g : \mathbb{R}^n \to \mathbb{R}^{n \times m}$ are (partially) unknown locally Lipschitz continuous functions that represent the drift dynamics and the input matrix, respectively. Eq. (1) represents a broad class of physical systems. Though our method can be straightforwardly extended to the more general case $\dot{x} = f(x, u)$, this structure significantly facilitates the computation of safe control inputs, as described in Section 2.1. We further assume that $\mathcal{X}$ is an open and connected set and that $\mathcal{U}$ is compact. When a locally Lipschitz controller

$\pi : \mathbb{R}^n \to \mathbb{R}^m$ is used, we can define the closed-loop system:

$$\dot{x} = f(x) + g(x)\pi(x). \qquad (2)$$

We also assume to know an upper bound $L_{\dot{x}} \in \mathbb{R}_+$ for the time-derivative of the dynamics, as specified by the following assumption.

**Assumption 2.1.** There exists a known positive constant $L_{\dot{x}}$, such that $\|\dot{x}\|_2 \leq L_{\dot{x}}$ holds for all $x \in \mathcal{X}$ and $u \in \mathcal{U}$.

We define safety by considering a *safe set* $\mathcal{C}$, as specified in the following.

**Definition 2.2** (Safety). The closed-loop system (2) is said to be safe (with respect to $\mathcal{C}$) if $x(0) \in \mathcal{C}$ implies $x(t) \in \mathcal{C}$ for all $t \geq 0$.

We consider the case where the safe set $\mathcal{C}$ corresponds to the superlevel set of some known continuously differentiable function $h : \mathbb{R}^n \to \mathbb{R}$ with 0 a regular value [1]:

$$\mathcal{C} \triangleq \{x \in \mathbb{R}^n \mid h(x) \geq 0\}.$$

This specification of $\mathcal{C}$ allows us to achieve safety through the control barrier function framework, which we discuss in the following section. This paper considers a compact $\mathcal{C}$ and a Lipschitz continuous functions $h$, as stated in the following assumption.

**Assumption 2.3.** The safe set $\mathcal{C}$ is compact and the function $h$ admits a Lipschitz constant $L_h$, i.e., $|h(x) - h(x')| \leq L_h \|x - x'\|_2$ holds for all $x, x' \in \mathcal{X}$.

Given an arbitrary nominal controller $\pi_{\text{nom}} : \mathcal{X} \to \mathcal{U}$, we aim to design a control law $\pi$ that renders the closed loop

---

[1] A function $h$ has 0 as a ***regular value*** if $h(x) = 0 \implies \frac{\partial h}{\partial x}(x) \neq 0$.

system (2) safe while following $\pi_{\text{nom}}$ as closely as possible. We make the following assumption regarding the nominal controller:

**Assumption 2.4.** The nominal controller $\pi_{\text{nom}}$ is locally Lipschitz in $\mathcal{X}$.

Assumption 2.4 is not restrictive, as it includes a broad class of controllers, including most types of neural networks.

### 2.1. Safety through Control Barrier Functions

Our method attempts to ensure safety using control barrier functions, which we introduce in the following.

**Definition 2.5** (Control Barrier Function (CBF) (Ames et al., 2017)). Let $\mathcal{C} \subset \mathcal{X}$ be the 0-superlevel set of a continuously differentiable function $h : \mathbb{R}^n \to \mathbb{R}$ with 0 a regular value. Furthermore, let $\alpha$ be a radially unbounded, strictly monotonically increasing function with $\alpha(0) = 0$. The function $h$ is a control barrier function (CBF) for (1) on $\mathcal{C}$ if, for all $x \in \mathcal{X}$,

$$\sup_{u \in \mathcal{U}} \dot{h}(x, u) \triangleq \sup_{u \in \mathcal{U}} \frac{\partial h}{\partial x}(x)\left(f(x) + g(x)u\right) > -\alpha(h(x)). \tag{3}$$

Intuitively, (3) states that there exists a control input that decreases the time derivative of the CBF as the boundary of the safe set $\mathcal{C}$ is approached, effectively slowing down the system as the boundary gets closer. If $f$ and $g$ were known perfectly, we could leverage this property to compute certifiably safe control inputs by including a CBF increase condition as a constraint when synthesizing control inputs:

$$\pi(x) = \underset{u \in \mathcal{U}}{\operatorname{argmin}} \quad \|u - \pi_{\text{nom}}(x)\|_2 \tag{4}$$
$$\text{s.t.} \quad \dot{h}(x, u) \geq -\alpha(h(x)).$$

Given the affine form of this constraint, this safety filter is a quadratic program (QP) that has a closed form solution (Ames et al., 2017) and can be solved fast enough for online safe controller synthesis (Gurriet et al., 2018).

Since we are dealing with unknown systems (3) is not directly applicable, and we need to formulate a robust CBF constraint that accounts for potential modeling errors. To this end, we assume the CBF constraint is feasible up to a positive margin $\epsilon > 0$:

**Assumption 2.6** (Robust CBF Feasibility). There exists a radially unbounded, strictly monotonically increasing function $\alpha$ with $\alpha(0) = 0$ and a positive scalar $\epsilon > 0$, such that

$$\sup_{u \in \mathcal{U}} \frac{\partial h}{\partial x}(x)\left(f(x) + g(x)u\right) \geq -\alpha(h(x)) + \epsilon \tag{5}$$

holds for all $x \in \mathcal{X}$.

Given the existence of a CBF, we note that this assumption is not very conservative. Intuitively, it states that we can violate the non-robust CBF by a margin of at most $\epsilon$.

### 2.2. Gaussian Processes and Reproducing Kernel Hilbert Spaces

We now introduce the tools used to model the the unknown functions $f$ and $g$.

Consider a prior model of the system $\hat{f}$ and $\hat{g}$, which can be set to zero if no prior knowledge is available. For the remainder of this paper, we assume $\hat{f} = \hat{g} = 0$ for simplicity. We then model the residuals $f - \hat{f}$ and $g - \hat{g}$ jointly by using a GP model. To obtain theoretical guarantees on the growth of the model error as new data is added, we assume that the residual model belongs to a reproducing kernel Hilbert space (RKHS). This is codified later in this section. In the following, we review both GPs and RKHSs, and provide some preliminary theoretical results.

A GP is a collection of random variables, of which any finite number is jointly Gaussian distributed. GPs are fully specified by a prior mean, which we set to zero without loss of generality, and kernel $k : \mathcal{X} \times \mathcal{X} \to \mathbb{R}$. We model each dimension of the unknown dynamics $f(x) + g(x)u$ using a separate GP. This is achieved by employing composite kernels

$$k_i(\mathbf{z}, \mathbf{z}') \triangleq k_{f_i}(x, x') + \sum_{j=1}^{m} u_j k_{g_{i,j}}(x, x')u_j', \tag{6}$$

where $\mathbf{z} \triangleq (x^\top, u^\top)^\top$. The kernels $k_{f_i}$ and $k_{g_{i,j}}$ capture the behavior of the individual entries of the residuals $f$ and $g$, respectively. In this paper, we do not require a specific form of kernel $k_{f_i}$ and $k_{g_{i,j}}$, though the choice of kernel significantly affects performance. We employ a composite kernel (6) because this allows us to leverage measurements of $\dot{x}$ to improve our learned model.

Consider $N$ noisy measurements $\mathcal{D}_{i,N} \triangleq \{\mathbf{z}^{(q)}, y_i^{(q)}\}_{q=1,\ldots,N}$ of the $i$-th entry of the time-derivative of the state, where

$$y_i^{(q)} = f_i(x^{(q)}) + \sum_{j=1}^{m} g_{i,j}(x^{(q)})u_j^{(q)} + \xi_i^{(q)} \tag{7}$$

and the measurement noise $\xi^{(q)}$ satisfies the following assumption.

**Assumption 2.7.** The measurement noise $\xi^{(q)}$ is iid $\sigma_{i,\text{ns}}$-sub Gaussian for every $i \in \{1, \ldots, n\}$.

The posterior of the model at an arbitrary state $\mathbf{z}^*$ is then

given by

$$\mu_{i,N}(\mathbf{z}^*) \triangleq \mathbf{k}_{i,*}^\top \left(\mathbf{K}_{i,N} + \sigma_{i,\mathrm{ns}}^2 \mathbf{I}_N\right)^{-1} \mathbf{y}_i,$$
$$\sigma_{i,N}^2(\mathbf{z}^*) \triangleq k_i(\mathbf{z}^*, \mathbf{z}^*) - \mathbf{k}_{i,*}^\top \left(\mathbf{K}_{i,N} + \sigma_{i,\mathrm{ns}}^2 \mathbf{I}_N\right)^{-1} \mathbf{k}_{i,*},$$

where $\mathbf{k}_{i,*} \triangleq (k_i(\mathbf{z}^*, \mathbf{z}^{(1)}), ..., k_i(\mathbf{z}^*, \mathbf{z}^{(N)}))^\top$, and the entries of the covariance matrix are given by $[\mathbf{K}_{i,N}]_{pq} = k_i(\mathbf{z}^{(p)}, \mathbf{z}^{(q)})$. In our approach, we employ $\mu_{i,N}$ as a model of our system, and $\sigma_{i,N}$ as a measure of uncertainty, which we employ to inform the data-collection trigger.

In the following, we refer to the mean and covariance matrix of the full multivariate Gaussian process model as

$$\boldsymbol{\mu}_N(\cdot) \triangleq (\mu_{1,N}(\cdot), \ldots, \mu_{n,N}(\cdot))^\top,$$
$$\boldsymbol{\Sigma}_N^2(\cdot) \triangleq \mathrm{diag}(\sigma_{1,N}^2(\cdot), \ldots, \sigma_{n,N}^2(\cdot)). \tag{8}$$

Furthermore, we employ $\mathcal{D}_N = \{\mathbf{z}^{(q)}, \mathbf{y}^{(q)}\}_{q=1,\ldots,N}$, where $\mathbf{y}^{(q)} \triangleq (y_1^{(q)}, ..., y_n^{(q)})^\top$, to refer to the full data set corresponding to $N$ system measurements.

To be able to ensure safety, we need to estimate how data affects the worst-case model error. Though this is impossible in a general setting, an estimate is possible if we assume that the entries of $f$ and $g$ belong to an RKHS, a very rich function space specified by the composite kernel (6).

**Assumption 2.8.** For every $i = 1, .., n$, the $i$-th entry of $f(x) + g(x)u$ belongs to the RKHS with reproducing kernel $k_i$, and the corresponding RKHS norm is bounded by a known positive scalar $B_i \in (0, \infty)$.

The satisfaction of Assumption 2.8 hinges on choosing the kernels correctly. In practice, if little is known about the system, then so-called "universal kernels" (e.g. Matérn or squared-exponential kernels) are often employed, which can approximate continuous functions arbitrarily accurately (Micchelli et al., 2006). Furthermore, there exist several techniques to mitigate kernel specification that can easily be applied to the methods described in this paper (Berkenkamp et al., 2019; Capone et al., 2022; Fiedler et al., 2021).

## 3. Safe Control Through Bandit Exploration

In this section, we present our main algorithm and theoretical guarantees.

Our algorithms employs two key quantities: the *lower confidence bound* (LCB) and *upper confidence bound* (UCB) of the time derivative of $h$ given the data. These quantities are

computed as

$$LCB_N(x, u) = \frac{\partial h}{\partial x}(x)\boldsymbol{\mu}_N(x, u) - L_h \beta_N \sqrt{\mathrm{tr}\left(\boldsymbol{\Sigma}_N(x, u)\right)} \tag{9}$$

$$UCB_N(x, u) = \frac{\partial h}{\partial x}(x)\boldsymbol{\mu}_N(x, u) + L_h \beta_N \sqrt{\mathrm{tr}\left(\boldsymbol{\Sigma}_N(x, u)\right)}, \tag{10}$$

where $L_h > 0$ is the global Lipschitz constant of $h$, specified in Assumption 2.3 and

$$\beta_N \triangleq \max_{i \leq n} \left(B_i + \sigma_{i,\mathrm{ns}}\sqrt{2\left(\gamma_{i,N} + 1 + \log\left(n\delta^{-1}\right)\right)}\right). \tag{11}$$

Here, $\gamma_{i,N}$ denotes the maximal information gain after $N$ rounds of data collection, and is computed as

$$\gamma_{i,N} = \max_{\mathbf{z}^{(1)}, \ldots \mathbf{z}^{(N)}} \frac{1}{2} \log\left|\mathbf{I}_N + \sigma_{i,\mathrm{ns}}^{-1}\mathbf{K}_{i,N}\right|.$$

Intuitively, the LCB (9) and UCB (10) respectively correspond to the lowest and highest permissible value for $\dot{h}(x, u)$ given the model error bound.

Given the nominal controller $\pi_{\mathrm{nom}}$, we leverage the LCB (9) to compute robustly safe control inputs by solving the second-order cone program

$$\pi_{N,\mathrm{safe}}(x) = \underset{u \in \mathcal{U}}{\mathrm{argmin}} \quad \|u - \pi_{\mathrm{nom}}(x)\|_2$$
$$\mathrm{s.t.} \quad LCB_N(x, u) \geq -\alpha(h(x)) + \frac{\epsilon}{2}. \tag{12}$$

Note that, due to the nature of the composite kernels (6), $\boldsymbol{\mu}_N(x, u)$ is a linear function of $u$ and $\boldsymbol{\Sigma}_N(x, u)$ is a diagonal matrix whose entries are positive definite quadratic functions of $u$. Hence, (12) can be solved straightforwardly with conventional second-order cone program optimizers.

We include the term $\frac{\epsilon}{2}$ in the constraint in (12) because this keeps the system from getting too close to the boundary. This ensures that when (12) becomes infeasible, there is still some distance to go until it reaches the boundary of the safe set, leaving us with enough time to explore the system.

As long as the robust safety filter (12) is strictly feasible, the resulting control policies are locally Lipschitz continuous (Castañeda et al., 2022), yielding unique and well-behaved closed-loop system trajectories.

If (12) is feasible for all $x \in \mathcal{C}$, we can leverage standard results from CBF theory (Ames et al., 2017) to show that the resulting trajectory is safe. However, since (12) includes model uncertainty, it becomes potentially infeasible in regions of the state space where uncertainty is high. We address this by employing a bandit-type exploration algo-

rithm to acquire new data whenever infeasibility occurs. Our algorithm is designed to collect sufficiently informative data to render (12) feasible before exiting the safe set. We now describe the corresponding sampling scheme in detail.

Let $N$ denote the number of data collected up to the current time. If (12) is strictly feasible, it is used to compute control inputs. As soon as strict feasibility is impossible, we switch to our bandit exploration strategy. This corresponds to the first instance in time when

$$\max_{u \in \mathcal{U}} LCB_N(x, u) = -\alpha(h(x)) + \frac{\epsilon}{2} \qquad (13)$$

holds and can be easily verified using a quadratic program. We denote this point in time by $t_{N+1}$. At this point, we switch to an exploration-driven controller $\bar{\pi}_{N+1}$, which can be chosen freely, provided that it is locally Lipschitz continuous and satisfies $\bar{\pi}(x(t_{N+1})) = u^{(N+1)}$, where $u^{(N+1)}$ maximizes the upper confidence bound

$$u^{(N+1)} = \arg\max_{u \in \mathcal{U}} UCB_N(x(t_{N+1}), u). \qquad (14)$$

In our experiments in Section 5, we choose $\bar{\pi}(x) \equiv u^{(N+1)}$. Choosing a control input this way is very similar to bandit exploration algorithms, which aim to find the maximum of an unknown function. In our case, we wish to maximize the unknown time derivative of the CBF $\dot{h}(x, u)$ given the state $x$, which we know to satisfy Assumption 2.6. We apply $\bar{\pi}_{N+1}$ for a time length of $\Delta t$, where $\Delta t$ corresponds to the data sampling frequency and is specified a priori. We then collect a noisy measurement of the time-derivative of the system

$$\mathbf{y}^{(N+1)} = f(x(t_{N+1})) + g(x(t_{N+1}))u^{(N+1)} + \xi^{(N+1)}$$

and update the GP model with the measurement data pair $(\mathbf{z}^{(N+1)}, \mathbf{y}^{(N+1)})$, where

$$\mathbf{z}^{(N+1)} = \left( \left( x(t_{N+1}) \right)^{\top}, \left( u^{(N+1)} \right)^{\top} \right)^{\top}.$$

After a time of $\Delta t$ has passed, we check if the optimization problem (12) is strictly feasible. To this end, we check if

$$\max_{u \in \mathcal{U}} LCB_N(x, u) > -\alpha(h(x)) + \frac{\epsilon}{2} \qquad (15)$$

holds. If so, we apply $\pi_{N,\text{safe}}$. Otherwise, we repeat the sampling procedure. These steps are summarized in Algorithm 1.

*Remark* 3.1. The condition (13) specifies that exploration starts when strict infeasibility is not given, as opposed to waiting until infeasibility occurs. In general, it is not advisable to wait for infeasibility to start exploring, as this means that the control law is not well-defined, which can be

---

**Algorithm 1** Safe Control via On-the-Fly Bandit Exploration

**Input:** Sampling time $\Delta t$, GP prior, CBF $h$, class-$\mathcal{KL}$ function $\alpha$
1: Set EXPLORE = FALSE
2: **for** $t \in [0, \infty)$ **do**
3:     **if** EXPLORE==FALSE **then**
4:         **if** $\max_{u \in \mathcal{U}} LCB_N(x, u) > -\alpha(h(x)) + \frac{\epsilon}{2}$ **then**
5:             Solve (12) and apply $\pi_{N,\text{safe}}(x)$
6:         **else**
7:             Set EXPLORE=TRUE
8:             Set $N = N + 1$.
9:             Set $t_N = t$.
10:           Set $x^{(N)} = x$.
11:           Compute $u^{(N)}$ by solving (14).
12:         **end if**
13:     **end if**
14:     **if** EXPLORE==TRUE **then**
15:         **if** $t < t_N + \Delta t$ **then**
16:             Apply a locally Lipschitz controller $\bar{\pi}$ with $\bar{\pi}(x^{(N)}) = u^{(N)}$.
17:             Collect noisy measurement $\mathbf{y}^{(N)} = \dot{x}(t_N) - \hat{f}(x(t_N)) - \hat{g}(x(t_N))u^{(N)} + \xi^{(N)}$
18:         **else if** $t = t_N + \Delta t$ **then**
19:             Set EXPLORE=FALSE
20:             Set $\mathcal{D}_N = \mathcal{D}_{N-1} \cap \{\mathbf{z}^{(N)}, \mathbf{y}^{(N)}\}$ and update GP.
21:         **end if**
22:     **end if**
23: **end for**

---

detrimental to control.

We then show that, if we choose $\Delta t$ high enough, the closed-loop system under Algorithm 1 is safe with high probability.

**Theorem 3.2.** *Let Assumptions 2.1, 2.3, 2.4 and 2.6 to 2.8 hold. Choose $\delta \in [0, 1]$ and $\beta_N$ as in (11). Moreover, choose the sampling time as*

$$\Delta t > \frac{\epsilon}{L_\alpha L_h L_{\dot{x}} \Delta N_{max}}, \qquad (16)$$

*where $\Delta N_{max}$ satisfies*

$$\Delta N_{max} > \frac{32\beta^2_{\Delta N_{max}} L_h^2}{\epsilon^2 \log\left(1 + \sigma^{-2}_{i,ns}\right)} \sum_{i=1}^{n} \gamma_{i,\Delta N_{max}} \qquad (17)$$

*Let the initial state $x(0)$ be strictly within the safe set, such that $\alpha(h(x(0))) \geq \epsilon$. Then, with probability at least $1 - \delta$, the closed-loop system under the policy specified by Algorithm 1 is safe. Furthermore, the closed-loop stops collecting data after at most $\Delta N_{max}$ collected observations.*

The proof of Theorem 3.2 can be found in Appendix B.

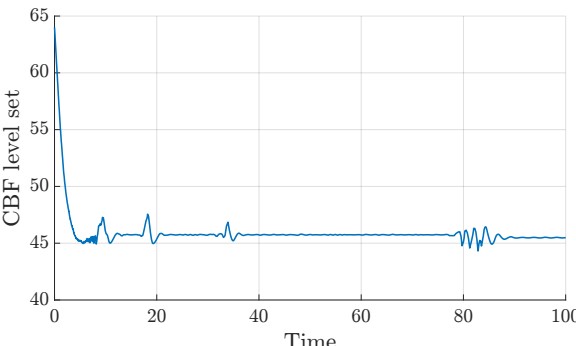

**(a)** Value of control barrier function $h(x)$ for cruise control example. Since no prior model is available for control, safety is obtained by efficiently learning a model online.

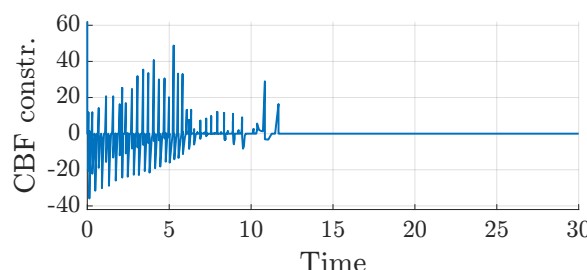

**(b)** Minimum of LCB of CBF time derivative for cruise control example. Spikes are due to training data set updates, leading to decreased model uncertainty. Positive values indicate infeasibility of the robust safety filter (12).

*Figure 2.* CBF and LCB for a single cruise control simulation using a sampling time of $\Delta t = 10^{-5}$s.

Intuitively, Theorem 3.2 states that if we choose the data collection frequency high enough, we guarantee safety after collecting a finite number of data. The quantity $\beta^2_{\Delta N_{\max}} \sum_{i=1}^n \gamma_{i,\Delta N_{\max}}$ is the only part in the right-hand side of (17) that depends on $\Delta N_{\max}$. It is closely related to the regret metric frequently found in bandit literature, which measures the deviation from optimal decisions. This quantity grows sublinearly for most commonly employed kernels, meaning there exists a $\Delta N_{\max}$ that satisfies (17). In practice, we can approximate by sampling the state and input space. Alternatively, we can obtain a (potentially crude) bound for this term based on the kernel hyperparameters. This is discussed in Appendix C.

## 4. Discussion and Limitations

Although we assume that the system structure (1) is affine in the control input, this is not strictly a requirement for theoretical guarantees and can be relaxed, provided that the optimization problems (12), (13) and (14) can be solved adequately.

Theorem 3.2 points at an interaction between various quantities, including the sampling frequency, model uncertainty, measurement noise, CBF violation tolerance, and Lipschitz constants of the system and CBF. In simple terms, Theorem 3.2 indicates that a higher sampling frequency is likely more beneficial for safety. However, other practical considerations can also improve safety. For instance, we can reduce the Lipschitz constant $L_{\dot{x}}$ of the system by cleverly restricting the permissible control input space $\mathcal{U}$, thereby reducing the bound (17).

A practical concern that may arise with control inputs $u$ geared towards exploration is that the corresponding inputs may strain the system and lead to undesirable behavior, e.g., if the input exhibits a high frequency and amplitude. However, computing the input by maximizing the UCB

(14) corresponds to choosing the optimistically safest input under uncertainty. Hence, it is reasonable to expect the corresponding input to be acceptable to the system, i.e., not damaging.

The primary focus of this work is the safety of the closed-loop function, and we do not address closed-loop control performance. However, the tools presented here can be straightforwardly extended to address performance, e.g., by designing a data collection scheme that aims to reduce model uncertainty whenever control performance is not satisfactory. Furthermore, our method can be potentially improved by preemptively exploring before infeasibility occurs, increasing the space of permissible control actions.

Although Assumption 2.6 can seem strict, it still offers flexibility, and there are ways to relax it in practice. Typically, only a portion of the safe set is visited during control. Hence, it is enough if the CBF is valid only for the corresponding subset of the state space. Furthermore, we can achieve conservatism by initially restricting the safe set to a region that is easy to control and then iteratively expanding it as more data is collected. This potentially allows us to gradually improve the CBF as more system knowledge allows us to expand the safe set. Alternatively, we can include conservatism in the safety requirement by computing the CBF condition from a collection of (potentially valid) CBFs and taking the worst case.

## 5. Numerical Experiments

We now showcase how our approach performs using two numerical simulations: a cruise control system and a quadrotor with ground dynamics. We assume to have either no prior model (cruise control example) or to know only the model component corresponding to the time derivatives (quadrotor). Note that this is insufficient to implement any state-of-the-art approach, as the prior model does not include

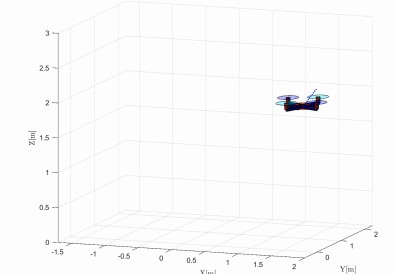

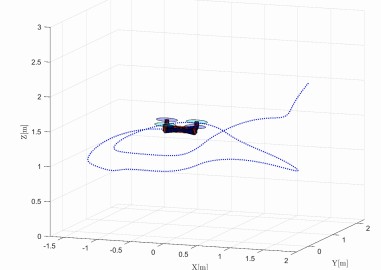

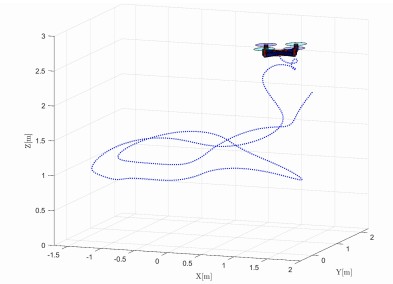

**(a)** The quadrotor starts at a state within the safe region where (12) is infeasible.

**(b)** Our algorithm computes exploratory control inputs without leaving the safe set. The observations are used to update the model. This phase corresponds roughly to the first two seconds of the simulation.

**(c)** After a finite amount of data is collected without exiting the safe set, the uncertainty is small enough to achieve the feasibility of the safety filter (12). This is expected from Theorem 3.2.

*Figure 3.* Visualization of a single trajectory of the quadrotor experiment using a sampling time of $\Delta t = 10^{-5}$s.

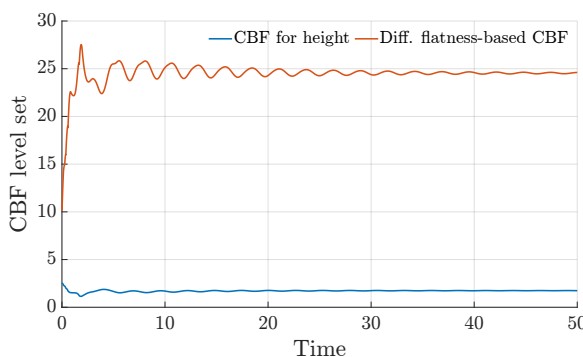

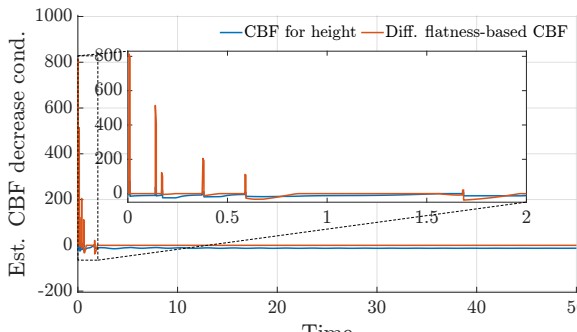

**(a)** Value of control barrier functions for quadrotor example.

**(b)** Minimum of the LCB of the CBF time derivative for quadrotor example.

*Figure 4.* CBF and LCB for a single quadrotor run using a sampling time of $\Delta t = 10^{-5}$s.

the influence of the control input. We additionally perform experiments that aim to answer the following questions:

- How does our method compare with random exploration during infeasibility?

- How does the choice of sampling time $\Delta t$ affect safety?

### 5.1. Cruise Control

We employ our approach to learn the road vehicle model presented in Castañeda et al. (2021) while applying an adaptive cruise control system. The states are given by $x = [v \ z]^\top$, where $z$ is the distance between the ego vehicle and the target vehicle in front, and $v$ denotes the ego vehicle speed. Our setup is fully described in Appendix A.1.

As a control barrier function, we employ $h(x) = z - T_h v$, where $T_h = 1.8$, which aims to maintain a safe distance between the ego vehicle and the vehicle in front. The nominal controller $\pi_{\text{nom}}(x)$ used for the robust CBF-SOCP (12) is a P-controller $\pi_{\text{nom}} = -10(v - v_d)$, where $v_d = 24$

corresponds to the desired velocity. We employ squared-exponential kernels to model $f$ and $g$ and assume to have $N = 10$ data points at the start of the simulation, which we employ exclusively to learn the kernel hyperparameters. We do so by minimizing the posterior likelihood (Rasmussen & Williams, 2006).

We simulate the system for 100 seconds. The CBF value for a single simulation run using a sampling time of $\Delta t = 10^{-5}$s is shown in Figure 2a. The minimum of the LCB is depicted in Figure 2b. The CBF value $h(x)$ is always above zero, meaning the system is safe. This is to be expected from Theorem 3.2. As can be seen in Figure 2b, the minimum of the LCB is frequently positive in the beginning of the simulation. This means that the safety filter (12) is infeasible during these time instances, particularly in the beginning. This leads to a high rate of exploratory inputs, which suddenly decreases model uncertainty, causing the LCB to spike. After approximately 6 seconds, enough information was collected to recover feasibility, after which a safe input can be obtained without further exploration.

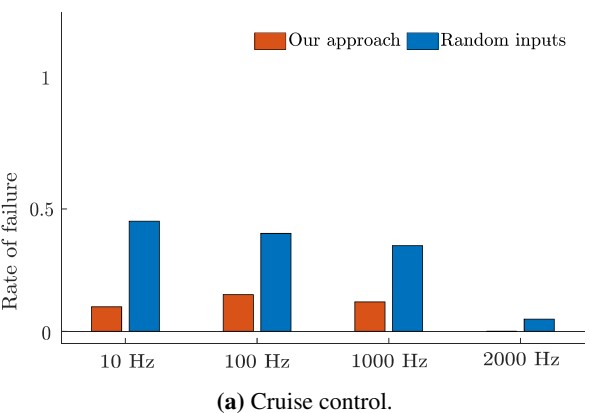
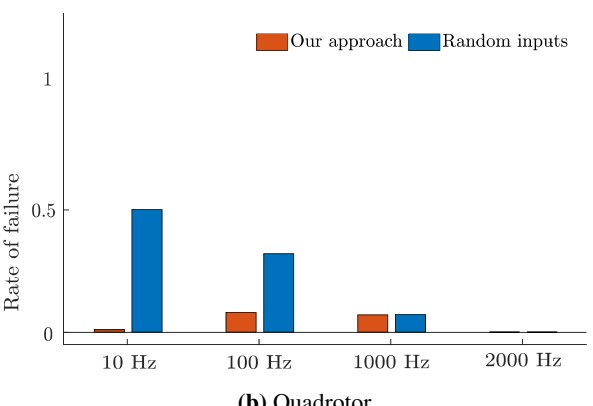

**(a)** Cruise control.      **(b)** Quadrotor.

*Figure 5.* Rate of failure using our approach and random inputs for the cruise control (left) and quadrotor (right) settings at varying sampling frequencies $(\Delta t)^{-1}$.

## 5.2. Quadrotor

We additionally showcase our approach using a numerical simulation of a quadrotor with ground dynamics. The quadrotor dynamics are described in detail in Appendix A.2.

The quadrotor states are given by $x = [p, v, q]$, where where $p \in \mathbb{R}^3$ is the global position, $v \in \mathbb{R}^3$ the global velocity, and $q \in \mathbb{R}^4$ is the system orientation in quaternion format. The control inputs $u = [T\ \omega]^\top$ consist of the thrust acceleration $T$ and the body-frame angle rates $\omega \in SO(3)$.

In this example, we employ two CBFs. The first is $h(x) = 10(p_z - T_z v_z)$, where $T_z = 0.1$, and is geared toward keeping altitude higher than zero. The second CBF utilizes the differential flatness of the quadrotor to restrict velocities, orientation we add a rotation, and the orientation of the thrust vector. It is fully described in Appendix A.2. When computing the exploratory control inputs (14), we alternate between CBFs. The nominal controller $\pi_{\text{nom}}(x)$ used for the robust safety filter (12) corresponds to a differentially flat controller, computed as in Faessler et al. (2018), and we consider bounded thrust, with $|T| \leq 15000$. Similarly to the cruise control setting, we use squared-exponential kernels and assume to have $N = 10$ data points at the start of the simulation to learn the kernel hyperparameters.

We simulate the system for $50$ seconds. The CBF value for a single run using a sampling time of $\Delta t = 10^{-5}$s can be seen in Figure 4a, the minimum of the LCB is depicted in Figure 4b. Figure 4 depicts the corresponding quadrotor trajectory. Similarly to the cruise control setting, infeasibility is frequently encountered initially, causing our algorithm to explore. After approximately 2 seconds, feasibility is recovered, causing the quadrotor to hover in place for the remainder of the simulation.

## 5.3. Comparison with Random Exploratory Control

Theorem 3.2 states that it is *sufficient* to collect data by applying (14) to the system in order to guarantee safety. However, other types of control inputs may also satisfy this requirement. In the following, we investigate how our approach performs compared to random exploratory control inputs. More specifically, we apply our method to the system with the following difference: instead of computing exploratory inputs by solving (14), we sample the control inputs from a uniform distribution on $\mathcal{U}$. We again consider the adaptive cruise control and quadrotor settings and investigate how our approach and the random input-based one perform using various sampling frequencies $\Delta t$. This is especially relevant, as many practical settings do not allow for arbitrarily high sampling frequencies.

We perform 100 simulations with different initial conditions, uniformly sampled from a region within the safe set. We report how often each method fails, i.e., leads to a positive value for the CBF during the simulation. The average number of failures for both settings is shown in Figure 5. A non-zero failure rate is expected at low sampling frequencies since too little data is collected to learn a model quickly. However, our approach nonetheless performs better than the random control input-based approach. This is because the inputs applied to the system are geared towards recovering the feasibility of the robust safety filter (12), whereas random inputs are not. We also observe that data efficiency is higher at higher sampling, leading to less collected data. This is because a higher sampling rate means that the elapsed time between data collection and model update is smaller, i.e., the model captures the true system more faithfully immediately after an update. This suggests that, if the maximal permissible sampling frequency is low, then particular effort should be put into updating the model as fast as possible.

## 6. Related Work

### 6.1. Control Barrier Functions

CBFs have been studied extensively in settings with perfectly known dynamics (Agrawal & Sreenath, 2017; Ames et al., 2017; Bansal et al., 2017; Wabersich & Zeilinger, 2018; Zeng et al., 2021). However, the setting where a known CBF is used to guide data collection when dynamics are unknown is an ongoing research question that has recently garnered increased attention. In (Choi et al., 2020; Taylor et al., 2020), the authors employ a neural network to learn the residual dynamics in a reinforcement learning setting. The safety of the learned policy is enforced by including a CBF constraint during training. An event-triggered approach for improving the learned dynamics model is proposed in (Lederer et al., 2024), where a CBF and the system structure are leveraged to guarantee safety.

### 6.2. Safe Exploration with Gaussian Processes

Gaussian processes have been used extensively in the context of safe control and exploration. The work of (Sukhija et al., 2023) leverages contextual bandits to safely tune a parametric controller. By formulating an upper confidence bound for the safety function, the method switches to safe parameters whenever the boundary of the safe set is reached. The works of Prajapat et al. (2022); Sui et al. (2015); Turchetta et al. (2016); Wachi & Sui (2020) consider exploration with finite state spaces, known dynamics, and unknown functions specifying safe state-action pairs. A lower confidence bound of the safety constraints ensures safe actions, whereas an upper confidence bound for the reward drives exploration. A significant difference between our method and those described above is that they explore safe states to expand the safe set and maximize return, whereas exploration in our approach is driven by the need to ensure safety. The methods of (Hewing et al., 2019; Koller et al., 2018) use a GP to formulate safety constraints over a finite horizon for a model predictive control algorithm.

## 7. Conclusion

We have presented a learning-based control approach for exploring on the fly to guarantee safety. Using a Gaussian process model, we have formulated a robust safety filter that guarantees safe inputs whenever feasible. Whenever infeasibility occurs we revert to a bandit exploration algorithm that is geared toward ensuring that feasibility is recovered before exiting the safe set. We show that if the sampling rate is chosen high enough, we can provably ensure safety with high probability. In future, we aim to apply the proposed approach to real-life and other complex systems.

## Acknowledgements

This work is supported by the DAAD programme Kon- rad Zuse Schools of Excellence in Artificial Intelligence (relAI), sponsored by the German Federal Ministry of Education and Research and by the European Research Council (ERC) Consolidator Grant "Safe data-driven control for human-centric systems (CO-MAN)" under grant agreement number 864686.

## Impact Statement

This paper presents work toward advancing the safety of Machine Learning-based methods. Safety plays a crucial role in developing Machine Learning techniques, as guaranteeing safety greatly enhances the applicability of existing methods. We feel that developing and further improving safe learning-based techniques such as the one presented in this paper will have a significant societal impact, as the spectrum of applications is substantial, including healthcare, autonomous driving, human-machine interaction, and various other robotics fields.

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

# A. Full Description of Numerical Experiments

## A.1. Cruise Control

In the following, we omit physical dimensions when describing the system model. The state space model of the cruise control is given by $\dot{x} = f(x) + g(x)u$, with state-dependent functions

$$f(x) = \begin{bmatrix} -\frac{1}{m}(\zeta_0 + \zeta_1 v + \zeta_2 v^2) \\ v_0 - v \end{bmatrix}, \quad g(x) = \begin{bmatrix} 0 \\ \frac{1}{m} \end{bmatrix} \tag{18}$$

and state $x = [v\ z]^\top$, where $z$ denotes the distance between the ego vehicle and the target vehicle in front, $v$ denotes the ego vehicle speed, $m = 1650$ its mass, and $\zeta_0 = 0.2$, $\zeta_1 = 10$, $\zeta_2 = 0.5$ are parameters that specify the rolling resistance.

## A.2. Quadrotor

The quadrotor dynamics are specified by the functions

$$\dot{p} = v, \qquad\qquad\qquad \dot{v} = g_{\mathrm{gr}}e_z + \zeta(p_z)Re_z T, \tag{19a}$$

$$\dot{R} = R[\omega]_\times \tag{19b}$$

where $p \in \mathbb{R}^3$ is the global position, $v \in \mathbb{R}^3$ the global velocity, and $R \in \mathrm{SO}(3)$ the system orientation. The parameter $g_{\mathrm{gr}} = 9.81$ denotes gravity, $[\,\cdot\,]_\times : \mathbb{R}^3 \to \mathbb{R}^{3\times 3}$ is the skew-symmetric mapping, and $e_z = [0\ 0\ 1]^\top$ is the unit-$z$ vector. The function $\zeta : \mathbb{R}_+ \to [0,1]$ models ground effects and takes the quadrotor height $p_z$ as an input variable. It is computed as Danjun et al. (2015)

$$\zeta(p_z) = 1 - \rho\left(\frac{r_{\mathrm{rot}}}{4p_z}\right)^2, \tag{20}$$

where $\rho = 5$ and $r_{\mathrm{rot}} = 0.09$ is the rotor radius.

The first CBF is $h(x) = 10(p_z - T_z v_z)$, where $T_z = 0.1$. It is geared toward keeping altitude higher than zero. The second CBF utilizes the differential flatness of the quadrotor to synthesize a viable safe set. In particular, we are concerned with restricting the system to safe positions defined as the 0-superlevel set of $h_p(p) = r^2 - \|p\|^2$. We extend $h_p$ to include velocities as $h_e(p,v) = \dot{h}_p(p,v) + \alpha h_p(p)$ for some $\alpha > 0$ to produce a relative degree-1 CBF for a double integrator system as in Nguyen & Sreenath (2016). To include orientation we add a rotation term to produce a CBF for the drone, $h(p,v,R) = h_e(p,v) - \lambda(1 - \frac{1}{2r}\frac{\partial h}{\partial p}Re_z)$, with some $\lambda \in (0, r^2/2)$ which shrinks the safe set to ensure that the thrust vector is pointing inwards whenever $h_e(p,v) = 0$.

# B. Proof of Theorem 3.2

We prove Theorem 3.2 in two steps. First, we show that the control inputs are well-behaved and the closed-loop system is safe if the robust CBF-SOCP (12) is strictly feasible. Afterward, we show that strict feasibility is always recovered before exiting the safe set when using our exploration approach.

We begin by leveraging Assumptions 2.7 and 2.8 to obtain a bound on the model error, which is crucial to establish safety.

**Lemma B.1.** *Let Assumptions 2.7 and 2.8 hold and let*

$$\beta_N \triangleq \max_{i \leq n}\left(B_i + \sigma_{i,ns}\sqrt{2\left(\gamma_{i,N} + 1 + \log\left(n\delta^{-1}\right)\right)}\right).$$

*Then, with probability at least $1 - \delta$,*

$$\|f(x^*) + g(x^*)u^* - \boldsymbol{\mu}(x^*, u^*)\|_2 \leq \beta_N\sqrt{tr\left(\boldsymbol{\Sigma}_N(x^*, u^*)\right)}$$

*holds for all $x \in \mathcal{X}$, $u \in \mathcal{U}$ and all $N \in \mathbb{N}$.*

*Proof.* By Capone & Hirche (2019, Lemma 1), with probability at least $1 - \delta$,

$$\left| f_i(x^*) + \sum_{j=1}^{m} g_{i,j}(x^*) u_j^* - \mu_{i,N}(x^*, u^*) \right| \leq \left( B_i + \sigma_{i,\text{ns}} \sqrt{2 \left( \gamma_{i,N} + 1 + \log \left( n\delta^{-1} \right) \right)} \right) \sigma_{i,N}(x^*, u^*)$$

holds for all $i = 1, ..., n$, all $x \in \mathcal{X}$, $u \in \mathcal{U}$, and all $N \in \mathbb{N}$. We then have

$$\| f(x^*) + g(x^*) u^* - \boldsymbol{\mu}(x^*, u^*) \|_2^2 \leq \sum_{i=1}^{n} \left| f_i(x^*) + \sum_{j=1}^{m} g_{i,j}(x^*) u_j^* - \mu_{i,N}(\mathbf{z}^*) \right|^2$$

$$\leq \sum_{i=1}^{n} \beta_{i,N}^2 \sigma_{i,N}^2(\mathbf{z}^*)$$

$$\leq \sum_{i=1}^{n} \beta_N^2 \sigma_{i,N}^2(\mathbf{z}^*)$$

$$= \beta_N^2 \operatorname{tr}\left( \boldsymbol{\Sigma}_N(x^*, u^*) \right).$$

$\square$

Lemma B.1 allows us to bound the GP model error at any given point $\mathbf{z}^*$ given any data with high probability. This plays a crucial role in establishing robust system safety in the face of model uncertainty.

**Safety under Strict Feasibility**

We first demonstrate that when the robust safety filter (12) is strictly feasible, the control law $\pi_{N,\text{safe}}$ ensures the system remains strictly within the safe set with high probability. Specifically, the control barrier function value of the closed-loop system increases over time whenever $\alpha(h(x)) < \frac{\epsilon}{2}$. To prove this statement formally, we first present some preliminary results.

**Lemma B.2.** *Let the robust safety filter (12) be strictly feasible for some $x \in \mathcal{X}$. Then the policy $\pi_{N,safe}$ specified by the safety filter (12) is Lipschitz continuous in a neighborhood of $x$.*

*Proof.* The proof is identical to that of Castañeda et al. (2022, Lemma 6). $\square$

Lemma B.2 is important to guarantee that the trajectory resulting from applying $\pi_{N,\text{safe}}$ is unique and well-behaved, allowing us to establish safety.

**Lemma B.3.** *Let Assumptions 2.1, 2.3 and 2.6 to 2.8 hold. Then, with probability at least $1 - \delta$, the following holds for all $x \in \mathcal{X}$, all $u \in \mathcal{U}$, and all $N \in \mathbb{N}$:*

$$LCB_N(x, u) \leq \dot{h}(x, u) \leq UCB_N(x, u). \tag{21}$$

*Proof.* We show the result only for the lower bound $LCB_N$ since the argument is identical for the upper bound $UCB_N$. By

Lemma B.1, with probability at least $1 - \delta$,

$$
\begin{aligned}
\dot{h}(x, u) &= \frac{\partial h}{\partial x}(x) \left( f(x) + g(x)u \right) \\
&= \frac{\partial h}{\partial x}(x) \left( \boldsymbol{\mu}_N(x, u) \right) + \frac{\partial h}{\partial x}(x) \left( f(x) + g(x)u - \boldsymbol{\mu}_N(x, u) \right) \\
&\geq \frac{\partial h}{\partial x}(x) \left( \boldsymbol{\mu}_N(x, u) \right) - \left\| \frac{\partial h}{\partial x}(x) \right\|_2 \left\| (f(x) + g(x)u - \boldsymbol{\mu}_N(x, u)) \right\|_2 \\
&\geq \frac{\partial h}{\partial x}(x) \left( \boldsymbol{\mu}_N(x, u) \right) - \left\| \frac{\partial h}{\partial x}(x) \right\|_2 \beta_N \sqrt{\operatorname{tr}\left( \boldsymbol{\Sigma}_N(x, u) \right)} + \frac{\epsilon}{2} \\
&\geq \frac{\partial h}{\partial x}(x) \left( \boldsymbol{\mu}_N(x, u) \right) - L_h \beta_N \sqrt{\operatorname{tr}\left( \boldsymbol{\Sigma}_N(x, u) \right)} \\
&= LCB_N(x, u).
\end{aligned}
$$

holds for all $x \in \mathcal{X}$, all $u \in \mathcal{U}$, and all $N \in \mathbb{N}$. $\qquad \square$

We additionally require the following result, which is a direct consequence of a classical result on CBFs (Ames et al., 2017, Theorem 1).

**Lemma B.4.** *For some $\tau_{max} \in \mathbb{R} \cup \infty$, let $\tilde{x}(t)$, $t \in [0, \tau_{max})$, be the unique and well-defined solution to the differential equation*

$$
\dot{\tilde{x}} = \tilde{f}(\tilde{x}), \quad \tilde{x} \in \mathcal{X}, \quad \tilde{x}(0) = \tilde{x}_0 \tag{22}
$$

*where $\tilde{f}$ is Lipschitz continuous around a neighborhood of $\tilde{x}(t)$ for all $t \in [0, \tau_{max})$. If $h(\tilde{x}(0)) > 0$ and*

$$
\frac{\partial h}{\partial \tilde{x}}(\tilde{x}(t)) \tilde{f}(\tilde{x}(t)) \geq -\alpha(h(\tilde{x}(t))) + \frac{\epsilon}{2} \tag{23}
$$

*holds for all $t \in [0, \tau_{\max})$, then $\alpha(h(\tilde{x}(t))) > 0$ for all $t \in [0, \tau_{\max})$.*

*Proof.* The proof is identical to that of Ames et al. (2017, Theorem 1). Note that, although Ames et al. (2017, Theorem 1) requires $\tilde{f}$ to be locally Lipschitz continuous in $\mathcal{X}$ and (23) to hold for any state in $\mathcal{X}$, these properties are only used to generalize the result to all solutions of (22) with arbitrary $\tilde{x}(0) \in \mathcal{X}$, which we do not require in this case. $\qquad \square$

We now show that finite-time trajectories obtained with the controller $\pi_{N,\text{safe}}$ are always strictly within the safe set, provided that the safety filter (12) is strictly feasible.

**Lemma B.5.** *Let Assumptions 2.1, 2.3 and 2.6 to 2.8 hold. For all $N \in \mathbb{N}$, define*

$$
\mathcal{X}_N = \left\{ x \in \mathcal{X} \;\middle|\; \max_{u \in \mathcal{U}} LCB_N(x, u) > -\alpha\left( h(x) \right) + \frac{\epsilon}{2} \right\}
$$

*and the dynamical system*

$$
\dot{x}_N = f(x_N) + g(x_N) \pi_{safe,N}(x_N), \quad t \in [t_N + \Delta t, t_{N+1}), \tag{24}
$$

*specified within the time interval $[t_N + \Delta t, t_{N+1})$, where the initial condition satisfies $x_N(t_N + \Delta t) \in \mathcal{X}_N$, $t_{N+1}$ is given by*

$$
t_{N+1} \triangleq \inf \left\{ t \geq t_N + \Delta t \;\middle|\; \max_{u \in \mathcal{U}} LCB_N(x, u) = -\alpha(h(x)) + \frac{\epsilon}{2} \right\}, \tag{25}
$$

*and $\alpha(h(x_N(t_N + \Delta t))) > 0$, and $\pi_{safe,N}$ is given by (12). Then, if $\mathcal{X}_N$ is non-empty, (24) has a unique solution $x_N(t)$. Furthermore, with probability at least $1 - \delta$, the trajectories $x_N(t)$ satisfy the following conditions for all $N \in \mathbb{N}$:*

*(i)* $x_N(t) \in \mathcal{C}$ *for* $t \in [t_N + \Delta t, t_{N+1})$.

*(ii) If* $\alpha\left(h(x_N(\hat{t}))\right) < \frac{\epsilon}{2}$ *for any* $\hat{t} \in [t_N + \Delta t, t_N)$, *then* $\alpha\left(h(x_N(t))\right) > \alpha\left(h(x_N(\hat{t}))\right)$ *for all* $t \in (\hat{t}, t_N)$.

*(iii) If* $\alpha\left(h(x_N(\hat{t}))\right) \leq \frac{\epsilon}{2}$ *for any* $\hat{t} \in [t_N + \Delta t, t_N]$, *then* $\alpha\left(h(x_N(t))\right) \geq \alpha\left(h(x_N(\hat{t}))\right)$ *for all* $t \in [\hat{t}, t_N]$.

*Proof.* We begin by showing condition (i), then show (ii) and (iii).

(i) Due to the definition of $\mathcal{X}_N$, the safety filter (12) is strictly feasible for all $x \in \mathcal{X}_N$. Furthermore, the definition of $t_{N+1}$ implies that a candidate solution $x_N(t)$ satisfies $x_N(t) \in \mathcal{X}_N$ for all $t \in [t_N + \Delta t, t_{N+1})$. Moreover, $\pi_{N,\text{safe}}$ is well-defined and strictly feasible in $\mathcal{X}_N$. By Lemma B.2, $\pi_{N,\text{safe}}$ is Lipschitz continuous around a neighborhood of any $x \in \mathcal{X}_N$. Since $f$ and $g$ are locally Lipschitz continuous, this implies that $f(x) + g(x)\pi_{N,\text{safe}}(x)$ is locally Lipschitz continuous in $\mathcal{X}_N$. Hence, by the Picard-Lindelöf Theorem, (24) has a unique solution $x_N(t)$. Since $\mathcal{X}_N$ is bounded and $x_N(t) \in \mathcal{X}_N$ for all $t \in [t_N + \Delta t, t_{N+1})$, $x_N(t)$ is also well-defined for all $t \in [t_N + \Delta t, t_{N+1})$. Moreover, due to the definition of $\pi_{N,\text{safe}}$, Lemma B.3 implies that, with probability at least $1 - \delta$,

$$\frac{\partial h}{\partial x}(x_N)\left(f(x_N) + g(x_N)\pi_{N,\text{safe}}(x_N)\right) \geq -\alpha(h(x_N)) + \frac{\epsilon}{2}$$

holds for all $N$ and all $x_N \in \mathcal{X}_N$. By Lemma B.4, $x_N(t) \in \mathcal{C}$ holds for all $t \in [t_N + \Delta t, t_{N+1})$ and all $N \in \mathbb{N}$ with probability at least $1 - \delta$.

(ii) The following holds with probability at least $1 - \delta$: For all $x_N(t)$ and all $N \in \mathbb{N}$, if $\alpha(h(x_N(t'))) < \frac{\epsilon}{2}$ for any $t' \in [t_N + \Delta t, t_{N+1})$, then, by Lemma B.3 and the definition of $\pi_{\text{safe},N}$:

$$\dot{h}(x, \pi_{\text{safe},N}(x_N(t'))) \geq -\alpha\left(h(x(t'))\right) + \frac{\epsilon}{2} > -\frac{\epsilon}{2} + \frac{\epsilon}{2} = 0$$

Due to continuity with respect to time, this means there exists an instance in time $t_c \in (t', t_{N+1})$ and an $\tilde{\varepsilon} > 0$, such that $\dot{h}(x_N(t), \pi_{\text{safe},N}(x_N(t))) > \tilde{\varepsilon}$ holds for all $t \in (t', t_c]$. By the Comparison Lemma (Khalil, 1996), $h(x_N(t)) > h(x_N(t')) + \tilde{\varepsilon}(t_c - t') > h(x_N(t'))$ for all $t \in (t', t_c]$. By the same argument, $h(x_N(t))$ is strictly increasing in $t$ whenever $h(x_N(t')) < h(x_N(t)) < \frac{\epsilon}{2}$, hence $h(x_N(t)) > h(x_N(t'))$ for all $t \in (t_c, t_N]$. Since $\alpha$ is strictly increasing, this implies the desired result.

(iii) The proof is identical to that of (ii), except that the strict inequality is replaced by a non-strict one. $\square$

A consequence of Lemma B.5 is that safety is guaranteed if infeasibility never occurs within the safe set:

**Corollary B.6.** *Let Assumptions 2.3, 2.4 and 2.6 to 2.8 hold. With probability* $1 - \delta$, *if*

$$\max_{u \in \mathcal{U}} LCB_N(x, u) > -\alpha(h(x)) + \frac{\epsilon}{2}, \quad \forall\, x \in \mathcal{C} \tag{26}$$

*holds for any* $N \in \mathbb{N}$, $\pi_{safe,N}(x)$ *is well-defined for all* $x \in \mathcal{C}$. *Furthermore,* $t_{N+1} = \infty$ *and the closed-loop system*

$$\dot{x}_N = f(x_N) + g(x_N)\pi_{N,safe}(x_N), \quad t \in [t_N + \Delta N, \infty),$$

*is safe.*

*Proof.* Due to the definition of $\pi_{\text{safe},N}$, (26) directly implies that its feasible region is non-empty, i.e., $\pi_{\text{safe},N}$ is well-defined, for all $x \in \mathcal{C}$. We now show that $t_{N+1} = \infty$ by contradiction. Hence, suppose $t_{N+1} < \infty$ holds. By Item (ii) in Lemma B.5, the trajectory $x_N(t)$ stays strictly in the interior of $\mathcal{C}$ for $t \in [t_N + \Delta N, t_{N+1})$. Due to continuity with respect to time, $x_N(t_{N+1}) = \lim_{t \to t_{N+1}} x_N(t) \in \mathcal{C}$. Recall that $t_{N+1}$ denotes the earliest instance when the CBF-SOCP specifying $\pi_{\text{safe},N}$ is no longer strictly feasible. Since $x_N(t) \in \mathcal{C}$ and $\pi_{\text{safe},N}$ is strictly feasible in $\mathcal{C}$, this is a contradiction, meaning $t_{N+1} = \infty$. $\square$

### Recovering Feasibility Through Bandit Exploration

We now analyze the effect of data collected under Algorithm 1 on the feasibility of the safety filter. Specifically, we demonstrate that by gathering enough data before exiting the safe set $\mathcal{C}$, strict feasibility is ensured anywhere in $\mathcal{C}$. To

achieve this, we will need the following preliminary results, which bound the growth of the cumulative worst-case model error.

**Lemma B.7.** *The posterior variance satisfies*

$$\sum_{q=1}^{N} \sigma_{i,q}^2 \left( \mathbf{z}^{(q-1)} \right) \leq \frac{2}{\log \left( 1 + \sigma_{i,ns}^{-2} \right)} \gamma_{i,N} \tag{27}$$

*for all $N$, all $i$ and all $\mathbf{z}^{(1)}, ..., \mathbf{z}^{(N)} \in \tilde{\mathcal{X}}$.*

*Proof.* Employing the same argument as in the proof of Srinivas et al. (2012, Lemma 5.4), we get

$$\sigma_{i,q}^2 \left( \mathbf{z}^{(q)} \right) \leq \frac{1}{\log \left( 1 + \sigma_{i,ns}^{-2} \right)} \log \left( 1 + \sigma_{i,ns}^{-2} \sigma_{i,q}^2 \left( \mathbf{z}^{(q)} \right) \right).$$

The result then follows from Srinivas et al. (2012, Lemma 5.3). □

**Lemma B.8.** *For all $N$, the posterior covariance matrices satisfy*

$$\sum_{q=1}^{N} tr \left( \mathbf{\Sigma}_N(x^{(N)}, u^{(N)}) \right) \leq \sum_{i=1}^{n} \frac{2}{\log \left( 1 + \sigma_{i,ns}^{-2} \right)} \gamma_{i,N}. \tag{28}$$

*Proof.* By Lemma B.7,

$$\sum_{q=1}^{N} tr \left( \mathbf{\Sigma}_N(x^{(N)}, u^{(N)}) \right) = \sum_{q=1}^{N} \sum_{i=1}^{n} \sigma_{i,N}^2 \left( \mathbf{z}^{(N-1)} \right) \leq \sum_{i=1}^{n} \frac{2}{\log \left( 1 + \sigma_{i,ns}^{-2} \right)} \gamma_{i,N}.$$

□

Lemma B.8 bounds the growth of the posterior covariance matrices.

We now show that strict feasibility in $\mathcal{C}$ is recovered after collecting at most $\Delta N_{\max}$ data points.

**Lemma B.9.** *Let Assumptions 2.3, 2.4 and 2.6 to 2.8 hold. The following holds with probability at least $1 - \delta$: If Algorithm 1 collects $\Delta N_{max} - 1$ measurements in $\mathcal{C}$, where $\Delta N_{max}$ satisfies*

$$\Delta N_{max} > \frac{32 \beta_{\Delta N_{max}}^2 L_h^2}{\epsilon^2 \log \left( 1 + \sigma_{i,ns}^{-2} \right)} \sum_{i=1}^{n} \gamma_{i,\Delta N_{max}}$$

*then the safety filter is strictly feasible everywhere in $\mathcal{C}$.*

*Proof.* We employ a proof by contradiction. Hence, assume that after $\Delta N_{\max} - 1$ observations have been collected, there exists an $x^c \in \mathcal{C}$, such that

$$\max_{u \in \mathcal{U}} LCB_{\Delta N_{\max}}(x^c, u) \leq -\alpha(h(x^c)) + \frac{\epsilon}{2}. \tag{29}$$

In the following, we set

$$x^{(\Delta N_{\max})} \triangleq x^c,$$
$$u^{(\Delta N_{\max})} \triangleq \arg \max_{u \in \mathcal{U}} LCB_{\Delta N_{\max}}(x^c, u).$$

Recall that Algorithm 1 only collects observations $x^{(N)}, u^{(N)}$ if (12) is not strictly feasible, i.e.,

$$LCB_N\left(x^{(N)}, u^{(N)}\right) \leq \max_{u \in \mathcal{U}} LCB_N\left(x^{(N)}, u\right)$$
$$\leq -\alpha\left(h(x^{(N)})\right) + \frac{\epsilon}{2}$$

for all $u \in \mathcal{U}$. Furthermore, due to Lemma B.3, with probability at least $1 - \delta$,

$$\sup_{u \in \mathcal{U}} \dot{h}\left(x^{(N)}, u\right) \leq \max_{u \in \mathcal{U}} UCB_N\left(x^{(N)}, u\right)$$
$$= UCB_N\left(x^{(N)}, u^{(N)}\right),$$

holds for all $u \in \mathcal{U}$ and all $N \in \mathbb{N}$. By employing the identity

$$LCB_N(x, u) - UCB_N(x, u) = -2L_h\beta_N\sqrt{\text{tr}\left(\boldsymbol{\Sigma}_N(x, u)\right)},$$

we then obtain, with probability at least $1 - \delta$, for all $N \in \mathbb{N}$:

$$-\alpha(h(x^{(N)})) + \frac{\epsilon}{2} \geq LCB_N\left(x^{(N)}, u^{(N)}\right)$$
$$\geq LCB_N\left(x^{(N)}, u^{(N)}\right) + \sup_{u \in \mathcal{U}} \dot{h}(x^{(N)}, u) - UCB_N\left(x^{(N)}, u^{(N)}\right)$$
$$= \sup_{u \in \mathcal{U}} \dot{h}(x^{(N)}, u) - 2L_h\beta_N\sqrt{\text{tr}\left(\boldsymbol{\Sigma}_N(x^{(N)}, u^{(N)})\right)}$$
$$\geq -\alpha(h(x^{(N)})) + \epsilon - 2L_h\beta_N\sqrt{\text{tr}\left(\boldsymbol{\Sigma}_N(x^{(N)}, u^{(N)})\right)},$$

where the last inequality is due to Assumption 2.6. Hence, with probability at least $1 - \delta$,

$$2L_h\beta_N\sqrt{\text{tr}\left(\boldsymbol{\Sigma}_N\left(x^{(N)}, u^{(N)}\right)\right)} \geq \frac{\epsilon}{2}, \quad \forall N \in \mathbb{N}.$$

By squaring and summing up from $N = 1$ to $\Delta N_{\max}$, then employing Lemma B.8 and (29), we get

$$\Delta N_{\max}\left(\frac{\epsilon}{2}\right)^2 = \sum_{N=1}^{\Delta N_{\max}}\left(\frac{\epsilon}{2}\right)^2$$
$$\leq \sum_{N=1}^{\Delta N_{\max}}\left(2L_h\beta_N\sqrt{\text{tr}\left(\boldsymbol{\Sigma}_N(x^{(N)}, u^{(N)})\right)}\right)^2$$
$$\leq \beta_{\Delta N_{\max}}^2 4L_h^2 \sum_{N=N_0}^{\Delta N_{\max}} \text{tr}\left(\boldsymbol{\Sigma}_N(x^{(N)}, u^{(N)})\right)$$
$$\leq \beta_{\Delta N_{\max}}^2 4L_h^2 \sum_{i=1}^{n} \frac{2}{\log\left(1 + \sigma_{i,\text{ns}}^{-2}\right)}\gamma_{i,\Delta N_{\max}}.$$

This implies

$$\Delta N_{\max} \leq \frac{32\beta_{\Delta N_{\max}}^2 L_h^2}{\epsilon^2 \log\left(1 + \sigma_{i,\text{ns}}^{-2}\right)} \sum_{i=1}^{n} \gamma_{i,\Delta N_{\max}},$$

which is a contradiction. Hence, the safety filter is strictly feasible in all of $\mathcal{C}$. $\qquad\square$

Next, we analyze the amount of data collected under Algorithm 1 before potentially exiting the safe set $\mathcal{C}$. Specifically, we show that the closed-loop system under Algorithm 1 collects at least $\Delta N_{\max}$ data points before exiting the safe set. To do so,

we will require the following result:

**Lemma B.10.** *Let Assumptions 2.1, 2.3, 2.4 and 2.6 to 2.8 hold. Then, with probability at least $1 - \delta$: consider the closed-loop system (2) and assume there exists a time interval $[t', t'']$ with $t' < t''$, such that*

$$
\begin{aligned}
\alpha(h(x(t'))) \leq &\frac{\epsilon}{2}, \\
\alpha(h(x(t))) <&\alpha(h(x(t'))), \quad \forall t \in (t', t''],
\end{aligned}
\tag{30}
$$

*and let*

$$
t_{N'} = \max_{N \in \mathbb{N}} t_N, \quad s.t. \quad t_N \leq t'
\tag{31}
$$

*denote the last time instance when data is collected up until $t'$. Then, the safety filter is not strictly feasible at times $t'$ and $t_N$, and*

$$
|\alpha(h(x(t_N))) - \alpha(h(x(t')))| < L_\alpha L_h L_{\dot{x}} \Delta t
$$

*holds, where $\Delta t$ is the sampling frequency used by Algorithm 1.*

*Proof.* We begin by showing that the safety filter is not strictly feasible at time $t'$ by contradiction. Hence, assume that the safety filter is strictly feasible at $t'$. Since $\alpha(h(x(t'))) \leq \frac{\epsilon}{2}$, by Item (iii) in Lemma B.5, $\alpha(h(x(t))) \geq \frac{\epsilon}{2}$ holds for some $t \in (t', t'']$. This contradicts (30).

We now show that strict feasibility does not hold at $t_{N'}$ by contradiction. Hence, assume strict feasibility holds at $t_{N'}$. Due to the definition of $t_{N'}$ (31), a new data point is not collected until strictly after $t'$. This implies strict feasibility at time $t'$, which is a contradiction.

Since the safety filter is not strictly feasible at time $t_{N'}$, and $t_{N'}$ is the last time instance when data is collected before $t'$, from the sampling frequency $\Delta t$, we have

$$
\Delta t = t_{N'+1} - t_{N'} > t' - t_{N'}.
$$

The result then follows from the Lipschitz continuity of $\alpha$, $x$ and $h$. $\square$

We now show that if the closed-loop system ever exits the safe set, it must collect at least $\Delta N_{\max}$ data points before doing so.

**Lemma B.11.** *Let Assumptions 2.1, 2.3, 2.4 and 2.6 to 2.8 hold and let $x(0)$ be strictly within the safe set, such that $\alpha(h(x(0))) \geq \epsilon$. Then the following holds with probability at least $1 - \delta$: if the closed-loop system under Algorithm 1 ever exits the safe set $\mathcal{C}$, then it collects at least $\Delta N_{max}$ data points before doing so.*

*Proof.* Assume that the closed-loop trajectory $x(t)$ reaches the boundary of the safe set $\mathcal{C}$ at some time $t_{\text{exit}}$ before exiting it:

$$
t_{\text{exit}} = \sup \left\{ \hat{t} \geq 0 \mid \alpha(h(x(t))) \geq 0 \, \forall \, t \in [0, \hat{t}] \right\}.
\tag{32}
$$

Due to the continuity of $x$, $\alpha$ and $h$, and $\alpha(h(x(0))) \geq \epsilon$, there exists a time $t_{\frac{\epsilon}{2}} < t_{\text{exit}}$, such that

$$
t_{\frac{\epsilon}{2}} = \sup \left\{ \hat{t} \geq 0 \mid \alpha(h(x(\hat{t}))) = \frac{\epsilon}{2}, \quad \alpha(h(x(t))) < \frac{\epsilon}{2}, \quad \forall t \in (\hat{t}, t_{\text{exit}}] \right\}.
\tag{33}
$$

Let $N_{\frac{\epsilon}{2}}$ and $N_{\text{exit}}$ denote the number of collected observations at time $t_{\frac{\epsilon}{2}}$ and $t_{\text{exit}}$, respectively:

$$
\begin{aligned}
N_{\frac{\epsilon}{2}} &= \arg\max_{N \in \mathbb{N}} t_N, \quad s.t. \quad t_N \leq t_{\frac{\epsilon}{2}} \\
N_{\text{exit}} &= \arg\max_{N \in \mathbb{N}} t_N, \quad s.t. \quad t_N \leq t_{\text{exit}}.
\end{aligned}
$$

Then, with probability at least $1 - \delta$: Due to Lemma B.10, we have

$$|\alpha(h(x(t_{N_{\frac{\epsilon}{2}}}))) - \alpha(h(x(t_{\frac{\epsilon}{2}})))| \leq L_\alpha L_h L_{\dot{x}} \Delta t \tag{34}$$

$$|\alpha(h(x(t_{N_{\text{exit}}}))) - \alpha(h(x(t_{\text{exit}})))| \leq L_\alpha L_h L_{\dot{x}} \Delta t. \tag{35}$$

We now analyze the growth of $\alpha(h(x(t)))$ between $t = t_N$ and $t = t_{N+1}$. We separately analyze the case where the safety filter is strictly feasible and the case where it is not. If the safety filter is not strictly feasible at time $t_{N-1}$, due to Assumption 2.1 and the sampling frequency $\Delta t$, the CBF changes between $t_{N-1}$ and $t_N$ at most by

$$\begin{aligned}
&|\alpha(h(x(t_N))) - \alpha(h(x(t_{N-1})))| \\
&\leq L_{\dot{x}} L_\alpha L_h |t_N - t_{N-1}| \\
&\leq L_{\dot{x}} L_\alpha L_h \Delta t.
\end{aligned} \tag{36}$$

By Item (iii) in Lemma B.5, if the safety filter is strictly feasible at time $t_{N-1}$ and $\alpha(h(x(t_{N-1}))) \leq \frac{\epsilon}{2}$, then $\alpha(h(x(t_N))) \geq \alpha(h(x(t_{N-1})))$. Hence, for all $t_{N-1} \in [t_{\frac{\epsilon}{2}}, t_{\text{exit}}]$, whether the safety filter is feasible or not,

$$\alpha(h(x(t_{N-1}))) - \alpha(h(x(t_N))) \leq L_{\dot{x}} L_\alpha L_h \Delta t \tag{37}$$

holds. We then have

$$\begin{aligned}
\frac{\epsilon}{2} &= \alpha(h(x(t_{\frac{\epsilon}{2}}))) - \alpha(h(x(t_{\text{exit}}))) \\
&= \alpha(h(x(t_{\frac{\epsilon}{2}}))) - \alpha(h(x(t_{\text{exit}}))) + \alpha(h(x(t_{N_{\text{exit}}}))) - \alpha(h(x(t_{N_{\frac{\epsilon}{2}}}))) - \alpha(h(x(t_{N_{\text{exit}}}))) + \alpha(h(x(t_{N_{\frac{\epsilon}{2}}}))) \\
&\leq \alpha(h(x(t_{N_{\frac{\epsilon}{2}}}))) - \alpha(h(x(t_{N_{\text{exit}}}))) + 2 L_\alpha L_h L_{\dot{x}} \Delta t \\
&= \sum_{N=N_{\frac{\epsilon}{2}}+1}^{N_{\text{exit}}} \alpha(h(x(t_{N-1}))) - \alpha(h(x(t_N))) + 2 L_\alpha L_h L_{\dot{x}} \Delta t \\
&\leq (N_{\text{exit}} - N_{\frac{\epsilon}{2}} - 1) L_\alpha L_h L_{\dot{x}} \Delta t + 2 L_\alpha L_h L_{\dot{x}} \Delta t \\
&= (N_{\text{exit}} + 1 - N_{\frac{\epsilon}{2}}) L_\alpha L_h L_{\dot{x}} \Delta t,
\end{aligned}$$

where the first inequality is due to (34), and the second inequality is due to (37). This implies

$$N_{\text{exit}} - N_{\frac{\epsilon}{2}} + 1 \geq \frac{\epsilon}{2 L_\alpha L_h L_{\dot{x}} \Delta t} > \Delta N_{\max},$$

where the last equality follows from the sampling frequency specification (16). Since $N_{\text{exit}}$, $N_{\frac{\epsilon}{2}}$ and $\Delta N_{\max}$ are integers, this implies that the amount of collected data satisfies $N_{\text{exit}} - N_{\frac{\epsilon}{2}} \geq \Delta N_{\max}$. $\qquad\square$

We now put everything together and prove Theorem 3.2.

*Proof of Theorem 3.2.* By Lemma B.11, Algorithm 1 collects at least $\Delta N_{\max}$ observations before ever exiting the safe set $\mathcal{C}$. Lemma B.9 states that the safety filter is strictly feasible within all of $\mathcal{C}$ after collecting at most $\Delta N_{\max}$ observations. Together with Corollary B.6 this implies that, with probability at least $1 - \delta$, the closed-loop system is safe and Algorithm 1 collects at most $\Delta N_{\max}$ observations.

$\qquad\square$

## C. Kernel-Dependent Bounds for $\Delta N_{\max}$

Theorem 3.2 specifies that

$$\Delta N_{\max} > \frac{32 \beta_{\Delta N_{\max}}^2 L_h^2}{\epsilon^2 \log\left(1 + \sigma_{i,\text{ns}}^{-2}\right)} \sum_{i=1}^n \gamma_{i,\Delta N_{\max}} \tag{38}$$

is a sufficient condition to guarantee safety. We now analyze how the right-hand side term behaves for different kernels. We begin by analyzing how the maximal information gain $\gamma_{i,N}$ depends on the maximal information gain of the kernels $k_{f_i}$ and $k_{g_{i,j}}$ used to model $f$ and $g$.

**Lemma C.1.** *Let $k_i$ be given as in* (6)*, and let $\gamma_{f_i,N}$ and $\gamma_{g_{i,j},N}$ denote the maximal information gain after $N$ rounds of data collection under the kernels $k_{f_i}$ and $k_{g_{i,j}}$, respectively. Then*

$$\gamma_{i,N} \leq \gamma_{f_i,N} + \sum_{j=1}^{m} \gamma_{g_{i,j},N} + 3m \log(N) \tag{39}$$

*Proof.* The proof follows directly the definition of the composite kernel $k_i$, Krause & Ong (2011, Theorem 2) and Krause & Ong (2011, Theorem 3). $\square$

We now dissect the right-hand side of (40). Define $\gamma_{\Delta N_{\max}} \triangleq \max_{i \leq n} \gamma_{i,\Delta N_{\max}}$ and $B \triangleq \max_{i \leq n} B_i$. From the definition of $\beta_{\Delta N_{\max}}$, it follows that

$$\beta_{\Delta N_{\max}}^2 \sum_{i=1}^{n} \gamma_{i,\Delta N_{\max}} \leq \beta_{\Delta N_{\max}}^2 n \gamma_{\Delta N_{\max}}$$

$$= n\gamma_{\Delta N_{\max}} B^2 + n\gamma_{\Delta N_{\max}} 2B\sigma_{i,\mathrm{ns}} \sqrt{2\left(\gamma_{\Delta N_{\max}} + 1 + \log\left(n\delta^{-1}\right)\right)} + n\gamma_{\Delta N_{\max}} \sigma_{i,\mathrm{ns}}^2 2\left(\gamma_{\Delta N_{\max}} + 1 + \log\left(n\delta^{-1}\right)\right)$$

$$\leq C_1(B, \sigma_{i,\mathrm{ns}}, n)\gamma_{\Delta N_{\max}} + C_{\frac{3}{2}}(B, \sigma_{i,\mathrm{ns}}, n)\gamma_{\Delta N_{\max}}^{\frac{3}{2}} + C_2(B, \sigma_{i,\mathrm{ns}}, n)\gamma_{\Delta N_{\max}}^2$$

$$\leq \left(C_1(B, \sigma_{i,\mathrm{ns}}, n) + C_{\frac{3}{2}}(B, \sigma_{i,\mathrm{ns}}, n) + C_2(B, \sigma_{i,\mathrm{ns}}, n)\right) \max\{\gamma_{\Delta N_{\max}}^2, 1\}$$

$$\leq \left(C_1(B, \sigma_{i,\mathrm{ns}}, n) + C_{\frac{3}{2}}(B, \sigma_{i,\mathrm{ns}}, n) + C_2(B, \sigma_{i,\mathrm{ns}}, n)\right)\left(\gamma_{\Delta N_{\max}}^2 + 1\right),$$

where

$$C_1(B, \sigma_{i,\mathrm{ns}}, n) = nB^2 + n2B\sigma_{i,\mathrm{ns}}\sqrt{2\left(1 + \log\left(n\delta^{-1}\right)\right)} + n\sigma_{i,\mathrm{ns}}^2 2\left(1 + \log\left(n\delta^{-1}\right)\right)$$

$$C_{\frac{3}{2}}(B, \sigma_{i,\mathrm{ns}}, n) = n2B\sigma_{i,\mathrm{ns}}\sqrt{2}$$

$$C_2(B, \sigma_{i,\mathrm{ns}}, n) = 2n\sigma_{i,\mathrm{ns}}^2.$$

For most commonly used kernels, the growth rate of the maximal information gain is of the form $\mathcal{O}(N^\omega \log(N))$, where $0 \leq \omega < 1$ (Srinivas et al., 2012). Hence, using Lemma C.1, we can bound $\gamma_N$ as $\gamma_N \leq C_\gamma N^\omega (\log(N))^\theta$ for some positive $C_\gamma$, $\omega$ and $\theta$, and $N$ large enough. Using an estimate of this type, we can compute a (potentially crude) upper bound for the right-hand side of (40) as

$$\frac{32\beta_{\Delta N_{\max}}^2 L_h^2}{\epsilon^2 \log\left(1 + \sigma_{i,\mathrm{ns}}^{-2}\right)} \sum_{i=1}^{n} \gamma_{i,\Delta N_{\max}} \leq C(B, \sigma_{i,\mathrm{ns}}, n)\left(C_\gamma \Delta N_{\max}^{2\omega}(\log(N))^\theta + 1\right) \leq \tilde{C}(B, \sigma_{i,\mathrm{ns}}, n)\Delta N_{\max}^{2\omega}(\log(N))^{2\theta} \tag{40}$$

where $C$ and $\tilde{C}$ are appropriate constants. This yields the sufficient condition for choosing $\Delta N_{\max}$

$$\Delta N_{\max} \geq \left(\tilde{C}(B, \sigma_{i,\mathrm{ns}}, n)(\log(\Delta N_{\max}))^{2\theta}\right)^{\frac{1}{1-2\omega}}, \tag{41}$$

which can easily be solved using numerical methods. For squared-exponential kernels, $\omega = 0$ and $\theta = n$. For linear kernels, $\omega = 0$ and $\theta = 1$, whereas for Matérn kernels, $\theta = 0$ and $\omega$ depends on its smoothness parameter (Srinivas et al., 2012).

