# OpenReview forum: "Learning Safe Control via On-the-Fly Bandit Exploration"
_ICML.cc/2025/Conference — ICML 2025 poster_

### Official Review · Reviewer_b8R8 · 2025-03-13

**Overall Recommendation:** 2

**Summary:**

This paper proposes a safe control method to address the problem of infeasible safety filter. The authors use Gaussian processes to learn the system dynamics. They use the lower confidence bound of the control barrier function (CBF) to identify the feasibility of the safety filter, and sample exploratory controls according to the upper confidence bound of CBF to reduce the model uncertainty when it is infeasible. They provide the guarantee of feasible safety filter under finite samples of online exploration, and evaluate the proposed safe control methods on numerical simulation of cruise control and quadrotor control.

**Claims And Evidence:**

The authors claim that the proposed safe control method is safe with high probability, and shows evidences via

1. theoretically proving the feasibility of safety filter within finite exploratory samples.

2. experimentally demonstrating the numerical results over the safe control of two different dynamical systems.

However, I think these evidences may not fully support the claim:

1. The author did not provide safety guarantee during the exploratory sampling procedure.

2. The default sampling frequency in the experiment of is 1e5 hz, which may not be practical in real-world experiments.

**Essential References Not Discussed:**

The main idea of this paper is to introduce an optimistic exploratory phase when pessimistic safety estimation is infeasible. I think an important work [1] is missing, which also introduces a UCB-based exploration strategy and online learns a backup policy to guarantee high-probability safety. I think the authors should include this paper in discussion.

[1] Sukhija, Bhavya, et al. "Gosafeopt: Scalable safe exploration for global optimization of dynamical systems." Artificial Intelligence 320 (2023): 103922.

**Experimental Designs Or Analyses:**

During the experiment, the authors track the value and LCB of the CBF function, and demonstrate the effectiveness of the proposed optimistic sampling method. They also show the control trajectories of the quadrotor.
I have following concerns regarding to the experiment part:

1. As mentioned above, the 1e5 sampling frequency may be too high for real-world deployment, Under low sample frequency the proposed method cannot guarantee high safe probability, as shown in Figure 5.

2. Some implementation details are missing (e.g. the setting of $\beta$，$\alpha$, and $\epsilon$).

**Methods And Evaluation Criteria:**

Overall I think the proposed method makes sense and benchmark used for evaluation are common-used embodiments in safe control.

**Other Comments Or Suggestions:**

1. Some typos: line 412: duplicated number. line 414: should be Figure 3?

**Other Strengths And Weaknesses:**

**Strength**

1. The addressed safe control problem without feasible safe filter is important.

**Weakness**

1. Theoretical and experimental concerns have been mentioned above.

2. The proposed method uses GP to jointly predict over the concatenation of states and controls, which may be difficult to scale to higher-dimensional systems.

**Questions For Authors:**

1. In Figure 5, seems the rate of failure under 10 hz is better than 100 hz and 1000 hz. Can you explain why this happens?

**Relation To Broader Scientific Literature:**

Existing safe controls requires that the safety filter is always feasible, or there is a backup policy to ensure safety during exploration. The authors propose to use UCB of the CBF to sample optimistically safe controls to learn the CBF to address the issue of infeasibility safety filter or unavailable backup policy. However, I think the provided theoretical and empirical results cannot fully support this claim.

**Theoretical Claims:**

As mentioned above, the absence of safety analysis during the exploration phase hinders the guarantee of high-probability safety. I think further analysis about the unsafe probability of using UCB to sample should be conducted.

---

> ### Author Rebuttal · Authors · 2025-04-01
>
> Thank you kindly for your review. We have corrected all typos and addressed all your major comments below. We want to stress at this point that our method does, in fact, guarantee safety, even throughout exploration.
>
> **Safety guarantee during exploration:** Our method does guarantee safety during exploration. This is, in fact, the whole point and one of the main contributions of our method. Our method explores in a way that recovers the feasibility of the CBF safety filter **before** the safe set is exited. In other words, the closed-loop system is always in the safe set during exploration. This is illustrated by Figure 1 and shown theoretically in Theorem 3.2. The key idea in the proof is to show that if we collect data with our exploration strategy quickly enough, we *must* recover feasibility before exiting the safe set. This is achieved by upper-bounding the information that can be gained inside the safe set, then showing that each collected data point contributes sufficiently toward reducing the information within the safe set (Lemma B.9). This is a crucial aspect of our method, and we will make this clearer in the revised manuscript by explicitly stating it in the abstract and introduction.
>
> **Specifications for experiments:** In the experiments, we set $\beta=10$, $\epsilon=0$ and $h$ equal to the identity function. The high choice of $\beta$ aims to avoid underestimating the model error. Note that $\beta=2$ is otherwise frequently chosen in the literature, e.g., in Berkenkamp et al. (2017) and Wachi et al. (2018). The choice of $\epsilon=0$ corresponds to assuming that the CBF constraint can be enforced robustly, whereas $h$ equal to the identity function is for simplicity.
>
> Berkenkamp, F., Turchetta, M., Schoellig, A., & Krause, A. (2017). Safe model-based reinforcement learning with stability guarantees. Advances in neural information processing systems, 30.
>
> Wachi, A., Sui, Y., Yue, Y., & Ono, M. (2018). Safe Exploration and Optimization of Constrained MDPs Using Gaussian Processes. Proceedings of the AAAI Conference on Artificial Intelligence, 32(1).
>
> **Sampling time:** Although the sampling time required to achieve safety consistently in the examples is high, note that they correspond to a setting with no prior knowledge. Hence, it is reasonable to expect a high sampling time, as otherwise, sufficient information cannot be collected before exiting the safe set.
>
> **GoSafeOpt paper:** Thank you for pointing out this paper to us. The paper does indeed employ similar tools and has a similar motivation. Some important differences include:
> - On a high level, the goal of GoSafeOpt is to expand a set that is already safe. Although our method can also be used for the same end, its main goal is to render the system safe under high levels of uncertainty. This is particularly useful for settings where other safety certificates can potentially fail.
> - A further difference that is related to the one above is that GoSafeOpt requires a set of initially safe control parameters at the beginning of exploration to guarantee safety. In contrast, our approach can guarantee safety purely through exploration, without requiring an initially safe set of control parameters.
> - In GoSafeOpt, the goal is to tune a parametric controller, and the GP-based safety conditions depend on the controller's parametric structure. In our setting, the GP model is independent of any underlying control structure.
> We will include this comparison in the revised paper.
>
> **Less safety violations at 10Hz:** We agree that this is somewhat unexpected since a higher sampling rate should yield a better model faster, which translates to safer control. However, it is important to remember that at lower sampling frequencies, the system can quickly reach the boundary of the safe set before more than a handful of data has been collected. In these settings, the exploratory input is applied for a considerable amount of time before potentially reverting to the safety filter-based control, making its role non-neglegible. Hence, we speculate that the exploratory control is particularly beneficial for safety in this particular instance.

---

### Official Review · Reviewer_23em · 2025-03-13

**Overall Recommendation:** 1

**Summary:**

## update after rebuttal
My evaluation of the paper has not changed after the rebuttal. The technical part of the paper is correct but does not bring much new insight on the topic. The assumption of known CBF for a system is stronger than existence of partial models and backup controllers. The experiments do not show particular benefits of using CBF, compared to reasonable designs of other exploration schemes.

The paper described a method for using control barrier functions to guide exploration under model uncertainty. Theoretical analysis on sample complexity is given in the bandit exploration setting. Experiments are shown on quadrotor and cruise control which have dynamics that are well-understood.

**Claims And Evidence:**

The claims on safety-guided bandit exploration are supported, modulo the limitation of experiments only on system that have well-understood dynamics with high controllability and limited uncertainty.

**Essential References Not Discussed:**

It should compare with methods that do use prior knowledge about the dynamics and backup controllers, because the need for a predefined CBF is implicitly imposing even stronger assumptions.

**Experimental Designs Or Analyses:**

See above.

**Methods And Evaluation Criteria:**

The main problem is although the paper claims to provably achieves safety "in a general setting that does not require any prior
model or backup controller" there is the requirement of a fully known control barrier function. This assumption is very much questionable, because the difficulty of coming up with CBFs are well-known even under fully specified dynamics models. The proposed framework can only be useful when the model uncertainty can be dominated by the known CBF and control authority, which is a setting that hardly requires the whole CBF mechanism in the first place. For instance, this assumption is equivalent to requiring a backup controller that's very easy to formulate. It is thus misleading to claim that the methods achieve safety in a general setting.

**Other Comments Or Suggestions:**

none

**Other Strengths And Weaknesses:**

none

**Questions For Authors:**

The main part of the paper says it is ok to assume prior model of the system $\hat f$ and $\hat g$ to be zero. How do you come up with the CBF as input to the system in that case? What happens if the collected data show that the learned dynamics violates the CBF that is assumed given? In the setting of the experiments, how are the assumption different from having a nominal understanding of the system dynamics and backup controller?

**Relation To Broader Scientific Literature:**

The assumptions about the systems make the need of CBF questionable. By only comparing with CBF-based methods, the paper is not well-positioned in the literature.

**Theoretical Claims:**

Under the assumptions the theoretical claims on sample complexity is reasonable.

---

> ### Author Rebuttal · Authors · 2025-04-01
>
> Thank you kindly for your review. We have addressed your comments and questions below.
>
>
>
> **Knowledge of CBF and difference to nominal understanding of the system dynamics/backup controller:** Although the CBF assumption cannot be ensured in a general setting, it still offers flexibility, and there are ways to relax it in practice. Typically, only a portion of the safe set is visited during control; hence, the CBF function only has to satisfy the corresponding conditions for the corresponding subset of the state space. Furthermore, we can achieve conservatism by initially restricting the safe set to a region that is easy to control and then iteratively expanding it as more data is collected. This potentially allows us to gradually improve the CBF as more system knowledge allows us to expand the safe set. Alternatively, we can include conservatism in the safety requirement by computing the CBF condition from a collection of (potentially valid) CBFs and taking the worst case.
>
> Additionally, we note that the setting where a CBF is known but the dynamics are unknown is an open ongoing research question that has garnered increased attention recently:
> - End-to-End Safe Reinforcement Learning through Barrier Functions for Safety-Critical Continuous Control Tasks (Cheng et al., 2019)
> - Reinforcement Learning for Safety-Critical Control under Model Uncertainty, using Control Lyapunov Functions and Control Barrier Functions (Choi et al., 2020)
> - Learning for safety-critical control with control barrier functions (Taylor et al., 2020)
> - Constraint-Guided Online Data Selection for Scalable Data-Driven Safety Filters in Uncertain Robotic Systems (Choi et al., 2023)
> - Safe Barrier-Constrained Control of Uncertain Systems via Event-triggered Learning (Lederer et al., 2024)
> - Learning-Based Prescribed-Time Safety for Control of Unknown Systems With Control Barrier Functions (Huang et al., 2024)
>
> Note that these works require additional assumptions beyond a known CBF to achieve safety.
>
> Furthermore, we believe our method still has value even in settings where the CBF is not strictly valid. One of our paper's main contributions is that we use exploration to recover safety if all other safety certificates fail. This strongly contrasts with other approaches, which either assume enough controllability to stay within the safe set at all times or require a safe backup controller. Hence, it provides a principled way to design an exploration-based controller that aims to achieve safety whenever other certificates become invalid. We will include this discussion in the revised paper.
>
>
>
> **Comparison with other baselines:** We are currently implementing a comparison with the heuristic proposed by (Choi et al., 2023), which computes control inputs by maximizing the LCB whenever infeasibility occurs instead of exploring the state space. We will also include a comparison with the baseline with a fully known model. We will report the results during the second round of reviews.
>
> **Designing a CBF with a zero model prior:** In general, it is impossible to design a CBF without any prior model knowledge. However, it is still less challenging than deriving an accurate system dynamics model, and this requirement can frequently be relaxed, as discussed above. Moreover, note that allowing a prior model of zero is not strictly an assumption of our method but rather something that it allows for, e.g., if we want to learn an unbiased model purely from data. Our paper highlights this because it contrasts with other methods, which require more assumptions/backup controllers even in settings where a prior model is available.
>
> **CBF design in experiments:** The CBF was taken from the cruise control setting from Ames et al. (2021). It restricts the velocity of the vehicle it comes closer to the vehicle in front. For the quadrotor, the first CBF (for avoiding ground collision) is similar to the cruise control setting, whereas the second CBF (for avoiding overrotation) corresponds to a quadratic equation that constrains the velocities and the orientation of the quadrotor.
>
> **What happens if the collected data show that the learned dynamics violates the CBF that is assumed given?** The CBF is not necessarily learned from the model. Hence, it can violate the learned dynamics. In fact, it can violate the dynamics under pessimism (this corresponds to the CBF LCB), which is when our method explores.
>
>
>
>
> Choi, J. J., Castaneda, F., Jung, W., Zhang, B., Tomlin, C. J., & Sreenath, K. (2023). Constraint-guided online data selection for scalable data-driven safety filters in uncertain robotic systems. arXiv preprint arXiv:2311.13824.
>
> Ames, A. D., Grizzle, J. W., & Tabuada, P. (2014, December). Control barrier function based quadratic programs with application to adaptive cruise control. In 53rd IEEE conference on decision and control (pp. 6271-6278). IEEE.

---

> > ### Comment · Reviewer_23em · 2025-04-04
> >
> > The rebuttal fails to address my primary concerns: while the results follow from standard derivations under the restrictive assumptions, the writing of the paper exaggerates their significance, and will very likely mislead practitioners and future research in this important area.
> >
> > The authors make the following claims in the abstract
> >
> > >By combining a safety filter with exploration in this manner, our method provably achieves safety in a general setting that does not require any prior model or backup controller, provided that the true system lies in a reproducing kernel Hilbert space.
> >
> > This very strong claim stands in sharp contrast with the Assumptions 2.1-2.8 that are made later on in the paper. While some of these assumptions are standard, the assumption of having a CBF is in most realistic cases way stronger assuming an approximate model or having a backup controller -- computing CBFs themselves can be extremely challenging and pretty much not doable in high-dimensions under completely precise models. The authors also said in the rebuttal
> >
> > >In general, it is impossible to design a CBF without any prior model knowledge.
> >
> > Then why would you make a statement in the paper that clearly is contradictory with this fact, which is well-known to the CBF community while clearly misleading to anyone who's unfamiliar with these techniques?
> >
> > We all agree that some approximation of the dynamics and CBF conditions can be allowed without hurting safety, both formally and practically. The technical part of the paper is one way of formalizing some aspect that has been folklore for a long time: If the assumptions are strong enough to ensure that the estimated system behaviors concentrate sufficiently under collected data, such that the confidence bounds guarantee safety with enough margin under a precomputed CBF over a nominal dynamics model and controller, then such explorations can be safe, simply following forward invariance. Although the use of GP and RKHS make the conditions on the sampling part more precise, they do not really bring new insights to the techniques or the problems.
> >
> > I'm happy to go into detailed discussions about the problems of accepting the paper as is.

---

> > > ### Author Response · Authors · 2025-04-08
> > >
> > > Thank you for your reply.
> > >
> > > **Claims:** We understand your concerns regarding our claims and how we phrase them. Naturally, our goal is not to mislead the community or anyone unfamiliar with control barrier functions. We are happy to revise our claims and provide more context on control barrier functions. In the revised paper, we will highlight the limitations of requiring a CBF and how to obtain them (e.g., via low-fidelity reduced order models (Molnar et al., 2024; Cohen et al. 2024) or from the learned model). We will reformulate the abstract and the sentence you pointed out in it as follows:
> > >
> > > *In this paper, we assume to know a control barrier function, which we combine with a learned model to specify a robust certificate that ensures safety if feasible [...] We show theoretically that combining a safety filter with bandit exploration in this manner achieves safety, provided the control barrier function is valid and the true system lies in a reproducing kernel Hilbert space.*
> > >
> > > Similarly, we will reformulate our claims in the introduction, main body of the paper, and conclusion to stress that a valid control barrier function is necessary to achieve our theoretical results. Furthermore, we will include a paragraph dedicated to our assumption's limitations in the discussion section, which will include the discussion from our previous rebuttal comment.
> > >
> > > Please note that our method does, in theory, allow for a zero prior model of the dynamics, which we think is an unusual and interesting result, particularly since it contrasts with existing literature on learning with CBFs, where a strictly non-zero model is required in addition to a CBF (Taylor et al., 2020; Choi et al., 2020; Lederer et al., 2024). However, we agree that, while this is a theoretically interesting feature, in most practical settings where a CBF is known, we will have some understanding of the dynamics, and this theoretical result should be presented in that light. With this in mind, we will align our claims in the abstract and introduction with the more commonly encountered setting of a known CBF with partially/superficially known dynamics. We will present the feature of not needing a prior model of the one-step dynamics from our theoretical result as a remark, accompanied by this discussion.
> > >
> > >
> > > **Further contributions:** We respectfully disagree with your latter statement:
> > >
> > > *The technical part of the paper is one way of formalizing some aspect that has been folklore for a long time: If the assumptions are strong enough to ensure that the estimated system behaviors concentrate sufficiently under collected data, such that the confidence bounds guarantee safety with enough margin under a precomputed CBF over a nominal dynamics model and controller, then such explorations can be safe, simply following forward invariance.  Although the use of GP and RKHS make the conditions on the sampling part more precise, they do not really bring new insights to the techniques or the problems.*
> > >
> > > Please note that our setting does not assume any additional CBF margin during exploration (i.e., we do not change the size of the safe set). Instead, the tightening of the constraint is used as a trigger such that whenever infeasibilities occur, our method explores to collect data in a way that recovers feasibility of the constraint and maintains safety.
> > >
> > > Additionally, while our work presents a valuable theoretical contribution in formalizing this "folklore", we also make an important methodological contribution in our sampling scheme: a significant part of our contribution is how data is collected here. In particular, an arbitrary data collection scheme is insufficient, as it does not guarantee that sufficient information is collected to make the CBF constraint valid again on time. This is where the bandit setup becomes critical, as it provides a framework to analyze which control inputs are most informative and will contribute most strongly towards maximizing the CBF time derivative. These techniques have not been used in the CBF literature and provide a novel way of addressing infeasibility of the CBF constraints.
> > >
> > > **Additional experiments:**
> > > As you suggested, we included a baseline with a backup controller (Choi et al., 2023). We tested their approach with a varying number of prior data. You can find the mean rate of success for their method below.
> > >
> > > | **Number of data**           | 0      | 5      | 10     | 20   | 50   | 200  | 500  | 1000 |
> > > |----|----|---|---|----|---|----|----|---|
> > > | **Success (safety) rate** | 0.0    | 0.0    | 0.0    | 0.3  | 0.4  | 0.5  | 0.9  | 0.9  |
> > >
> > >
> > > Cohen, M. H., Molnar, T. G., & Ames, A. D. (2024). Safety-critical control for autonomous systems: Control barrier functions via reduced-order models. Annual Reviews in Control, 57, 100947.
> > >
> > > Molnar, T. G., Cosner, R. K., Singletary, A. W., Ubellacker, W., & Ames, A. D. (2021). Model-free safety-critical control for robotic systems. IEEE robotics and automation letters, 7(2), 944-951.

---

### Official Review · Reviewer_kuHs · 2025-03-13

**Overall Recommendation:** 4

**Summary:**

The paper proposed an online safe control algorithm with Gaussian process models of the dynamics and bandit-type exploration to learn the dynamics. Then, the learned dynamics are combined with a control barrier function to ensure online safety. The control signal is solved with a safety filter with a lower confidential bound for robustness. The data collection requires a high frequency to prevent safety violations and infeasibility of the safety filter optimization.

## Update after rebuttal

My major concern is the connection to previous thoeritical papers about safe exploration, which is resovled by the rebuttal.

**Claims And Evidence:**

I examined all the claims in the paper, and all of them are supported by clean and convincing evidence.

**Essential References Not Discussed:**

The results are bandit-like exploration algorithm to ensure safe control with control barrier functions, but similar ideas have been explored in other safe exploration (without using control barrier functinos) literature like

**Experimental Designs Or Analyses:**

Like I said in Methods And Evaluation Criteria, The evaluation criteria might miss some baselines; now, it only includes random exploration. I suggest the authors to include [1] and its following work like [2] to compare the performance.

[1] Sui, Yanan, et al. "Safe exploration for optimization with Gaussian processes." International conference on machine learning. PMLR, 2015.

[2] Wachi, A., Sui, Y., Yue, Y., & Ono, M. (2018). Safe Exploration and Optimization of Constrained MDPs Using Gaussian Processes. Proceedings of the AAAI Conference on Artificial Intelligence, 32(1).

**Methods And Evaluation Criteria:**

The proposed method makes sense. The assumptions might be too strong, but it is a general problem of control barrier function-based methods, not the problem of authors.

The evaluation criteria might miss some baselines; now, it only includes random exploration. I suggest the authors to include [1] and its following work like [2] to compare the performance.

[1] Sui, Yanan, et al. "Safe exploration for optimization with Gaussian processes." International conference on machine learning. PMLR, 2015.

[2] Wachi, A., Sui, Y., Yue, Y., & Ono, M. (2018). Safe Exploration and Optimization of Constrained MDPs Using Gaussian Processes. Proceedings of the AAAI Conference on Artificial Intelligence, 32(1).

**Other Comments Or Suggestions:**

N/A. Good paper.

**Other Strengths And Weaknesses:**

# Strength
1. The authors explain a lot about the solution existence, problem difficulties, and limitations. I appreciate them very much.
2. The experimental results show the effectiveness of pthe roposed algorithm.

# Weakness
1. The required sampling time might not be realistic in the real world.
2. As the data accumulates, the dimension of the safety filter optimization problem will increase.

**Questions For Authors:**

1. Can you report the computation time in the experimental section?
2. Can you explain the difference between this paper and previous safe exploration with GP and bandit style in the following references?

If the differences are properly justified, I will further improve my score. Overall, it is a good paper.

[1] Sui, Yanan, et al. "Safe exploration for optimization with Gaussian processes." International conference on machine learning. PMLR, 2015.

[2] Wachi, A., Sui, Y., Yue, Y., & Ono, M. (2018). Safe Exploration and Optimization of Constrained MDPs Using Gaussian Processes. Proceedings of the AAAI Conference on Artificial Intelligence, 32(1).

[3] Prajapat, Manish, et al. "Near-optimal multi-agent learning for safe coverage control." Advances in Neural Information Processing Systems 35 (2022): 14998-15012.

**Relation To Broader Scientific Literature:**

The results are novel in the sense of combining control barrier functions, but similar ideas have been explored in other safe exploration literature like

[1] Sui, Yanan, et al. "Safe exploration for optimization with Gaussian processes." International conference on machine learning. PMLR, 2015.

[2] Wachi, A., Sui, Y., Yue, Y., & Ono, M. (2018). Safe Exploration and Optimization of Constrained MDPs Using Gaussian Processes. Proceedings of the AAAI Conference on Artificial Intelligence, 32(1).

[3] Prajapat, Manish, et al. "Near-optimal multi-agent learning for safe coverage control." Advances in Neural Information Processing Systems 35 (2022): 14998-15012.

**Theoretical Claims:**

I briefly checked the proof of Theorem 3.2, it looks good and corret to me. I didn't check the correctness line by line.

---

> ### Author Rebuttal · Authors · 2025-04-01
>
> Thank you kindly for reviewing our paper. Please find our answers to your questions and comments below.
>
> **Comparison with other references and baselines:** Thank you for suggesting the additional references. Although the methods of Sui et al. (2015), Wachi et al. (2018) and Prajapat et al. (2022) employ similar tools (the most notable being a combination of GP-based LCBs and UCBs for safety and optimistic exploration), there exist some fundamental differences between their methods and ours. Arguably the biggest difference on a conceptual level is that exploration in their approaches is driven by return maximization. In contrast, exploration in our approach is driven by the need to ensure safety. Moreover, their approaches do not allow the (pessimistically) unsafe states to be visited, whereas ours does, using exploration to reduce uncertainty while in those states. Other differences are as follows:
> - Their methods require a tabular MDP setting, which scales poorly with the number of state space dimensions and actions.
> - They are designed for a discrete-time setting, whereas our approach considers a continuous-time setting.
> - Their methods require the transition dynamics to be known perfectly, whereas our approach does not.
> - A further difference tied to the previous point is how optimistic exploration is used. The methods of Sui et al. (2015) and Wachi et al. (2018) utilize exploration to expand the safe set and maximize the returns under optimism. However, the safe set is expanded only by visiting safe states, i.e., by incrementally moving toward the unsafe states. Note that this is typically only achievable with perfectly known dynamics. In contrast, our approach expands the safe set by directly exploring the states that are unsafe under pessimism.
>
> After carefully analyzing the method and code of Wachi et al. (2018), we have concluded that to implement their method on one of our baselines, we would require significant changes to their method:
> - First, we would need to discretize the dynamical systems. While this is possible for the cruise control example, it is practically infeasible for the quadrotor. This is because the state space is 10-dimensional, and the nonlinear dynamics are strongly state-dependent, meaning that a dense graph is required. However, this implies a graph with 10^20 nodes for just 20 support points per dimension.
>
> - Secondly, we would need to address the fact that the dynamics are unknown in our setting. To this end, we could either use the GP mean to estimate the state-transition dynamics or a pessimistic estimate of the transition dynamics under model uncertainty and the pessimistic safety constraint. Moreover, the state transition graph would have to be updated after every time step, which was not intended under the original algorithm.
>
> Overall, we feel that directly implementing their code on our setting would entail design decisions that would significantly alter the original method, making a fair comparison difficult. However, we will thoroughly discuss these papers in the revised manuscript. Moreover, we will include a comparison with the heuristic proposed by Choi et al. (2023), which corresponds to an equation that attempts to maximize the LCB whenever infeasibility occurs.
>
> **Regarding computation time:** Our method mainly requires computation time for GP predictions and to solve the optimization problems involving the LCB (exploration) and UCB (safe control) of the CBF. Please note that, due to the nature of the CBF optimization problem, we only require a single GP pass to formulate the required vectors and matrices, after which we only have to solve a convex optimization problem. Hence, the GP computation scales with the data, whereas the CBF optimization does not. However, sparse inference tools can be easily applied to improve scalability. Below, we report the average computation time for each operation as a function of the data using vanilla GPs (we use the GPyTorch toolbox without GPU acceleration):
>
> | | GP  | Optim. |
> | ---- | ------ | ---- |
> | n=0 | 0.0399 | 0.0346 |
> | n=10| 0.0407 |  0.233 |
> | n=50| 0.0410 |  0.0174|
> | n=200| 0.0448 | 0.0194 |
> | n=500| 0.0654 | 0.0182 |
>
> The GP column corresponds to the average time of GP computations per iteration, which scales with the data, as expected. The Optim column corresponds to the average time taken by the optimization problem, which stays approximately constant with the amount of data.

---

> > ### Comment · Reviewer_kuHs · 2025-04-08
> >
> > Apologize for the late reply. Thanks for the clarifications, now the novelty are more clear to me. I will update my score.
> >
> > It will be helpful to include the discussion briefly in the revision.

---

### Official Review · Reviewer_kKqd · 2025-03-13

**Overall Recommendation:** 4

**Summary:**

The work presents an approach to design certifiably safe feedback controllers for a priori unknown systems. The authors propose using gaussian processes to approximate the learned model (or its error with respect to the true model). Then, they leverage bandit theory to propose upper and lower bounds to the model estimates, using the lower bounds for ensuring safety through a control barrier function and deciding when infeasibility occurs. Under infeasible conditions (when the controller is not able to ensure safety given the current model bound), the method uses the upper confidence bound to guide exploration and collect data to improve the model estimates. They provide theoretical results showing that under assumptions, the method ensures safety up to a given probability bound. They demonstrate the effectiveness of the work in several control problems.

**Claims And Evidence:**

The claims proposed are well supported, and the theoretical results and experiments justify the contributions proposed.

**Essential References Not Discussed:**

None.

**Experimental Designs Or Analyses:**

The experiment results are thorough and showcase the effectiveness of the work. I do have a question regarding the statement in the introduction on the given examples not being solvable by the current state of the art. The experiments are interesting, but I do not really understand how these are not solvable by current methods, given that they seem fairly standard control problems.

**Methods And Evaluation Criteria:**

The theoretical results are sensible, and the proposed experiments seem adequate for the method proposed.

**Other Comments Or Suggestions:**

See above and below. Overall, I find this a solid paper.

**Other Strengths And Weaknesses:**

Strengths:
- The paper has solid theoretical results, does not overstate the claims and showcases the method effectively.
- The idea of leveraging bandit bounds for model uncertainty decision making is interesting.

Weaknesses:
- Some (critical) statements are made and quickly dismissed, see below.
- The method still depends on critical knowledge or careful heuristics when picking the kernels, and requires having valid control barrier functions.

**Questions For Authors:**

- The main question I have is regarding the assumption that we do not know the system dynamics and we need to learn them, and yet we do know a control barrier function for the system. Is this realistic for most applications?
- The choice of the kernel (although it is discussed) is not obvious. Is there any disadvantage to choosing "universal kernels"? And what are their limitations?

**Relation To Broader Scientific Literature:**

The authors do a good job at relating the work to existing safe control literature. I do not have any comments on this.

**Theoretical Claims:**

I checked the presented proofs for the main theorem (and necessary lemmas presented in the appendix) to the best of my capacity (given the reviewing load). The results seem correct and the proofs were well structured. I did not find any issues.

---

> ### Author Rebuttal · Authors · 2025-04-01
>
> Thank you kindly for your review. Below, you will find our answers to your comments.
>
> **Regarding the claim of solvability of the benchmarks:** This claim is tied to the controller having a prior mean model of zero, no measurement data, and only a CBF to guide it. Although existing methods can effectively control these systems, the control design in existing works is informed by a prior (non-zero) model of the dynamics. Alternatively, a backup controller is required. Our approach only requires the CBF and a well-specified GP to inform the control design. We will reformulate the sentence and include this discussion in the revised paper.
>
> **The choice of kernel is not obvious:** We agree with this statement. Choosing appropriate kernels for modeling dynamical systems is generally a nontrivial task, and there exists an extensive body of literature on how to do so while mitigating model misspecification; see, e.g., Berkenkamp et al. (2019), Fiedler et al., (2021), Capone et al., (2022). Crucially, although many kernels can satisfy the requirements of our approach, some kernels can generalize far better than others, significantly reducing the amount of data required to achieve safety. For example, if the system has a linear component, it is typically beneficial to include a linear component in the kernel, and sometimes, a linear kernel can even outperform a universal kernel in this sense. We will include this discussion in the paper.
>
> **Knowing a CBF without knowing the dynamics:** While it is generally impossible to formulate a valid CBF without any form of model knowledge, doing so is easier than obtaining an actual system model, since the existence of a CBF typically implies the existence of (potentially infinitely) many others. Furthermore, the assumption that the CBF is valid can be relaxed in many ways: the CBF only needs to be valid in parts of the state space that are visited by the closed-loop system. Moreover, our approach can gradually expand the safe set, enabling incremental improvements to the CBF. We also note that the assumption that a CBF is known despite not having an accurate model is common in the literature; see, e.g., Cheng et al. (2019), Choi et al., (2020), Taylor et al., (2020).
>
> Berkenkamp, F., Schoellig, A. P., & Krause, A. (2019). No-regret Bayesian optimization with unknown hyperparameters. Journal of Machine Learning Research, 20(50), 1-24.
>
> Capone, A., Lederer, A., & Hirche, S. (2022, June). Gaussian process uniform error bounds with unknown hyperparameters for safety-critical applications. In International Conference on Machine Learning (pp. 2609-2624). PMLR.
>
> Cheng, Richard, Gábor Orosz, Richard M. Murray, and Joel W. Burdick. "End-to-end safe reinforcement learning through barrier functions for safety-critical continuous control tasks." In Proceedings of the AAAI conference on artificial intelligence, vol. 33, no. 01, pp. 3387-3395. 2019.
>
> Choi, J., Castañeda, F., Tomlin, C. J., & Sreenath, K. (2020, July). Reinforcement Learning for Safety-Critical Control under Model Uncertainty, using Control Lyapunov Functions and Control Barrier Functions. In Robotics: Science and Systems (RSS). 2020.
>
> Fiedler, C., Scherer, C. W., & Trimpe, S. (2021, May). Practical and rigorous uncertainty bounds for Gaussian process regression. In Proceedings of the AAAI conference on artificial intelligence (Vol. 35, No. 8, pp. 7439-7447).
>
> Taylor, A., Singletary, A., Yue, Y., & Ames, A. (2020, July). Learning for safety-critical control with control barrier functions. In Learning for dynamics and control (pp. 708-717). PMLR.

---

### Decision · Program_Chairs · 2025-05-01

**Decision:**

Accept (poster)

**Comment:**

This paper is squarely borderline but the AC argues for (weak) acceptance after weighing the reviews for strengths and weaknesses. Firstly, safety of ML-based control --particularly during exploration and data collection -- is an active area of research and bringing together the CBF methodology along with GPs to characterize system uncertainty is a coherent approach to attempt to advance further. The primary criticisms against the paper are that  (1)"the assumption of known CBF for a system is stronger than existence of partial models and backup controllers" and (2) the experiments are limited, shown for Quadrotor/Cruise systems with well understood dynamics and unclear implications for high-dimensional systems.

 That said, the proposed method explores in a way that recovers the feasibility of the CBF safety filter before the safe set is exited --  by upper-bounding the information that can be gained inside the safe set, then showing that each collected data point contributes sufficiently toward reducing the information within the safe set. This kind of argument is non-trivial -- requiring connecting dots between CBF synthesis, sample complexity and choice of function families to capture uncertain dynamics; hence the could be useful for future iterations of controllers that admit safe exploration.